# Measurement report: Source characteristics of water-soluble organic carbon in PM$_{2.5}$ at two sites in Japan, as assessed by long-term observation and stable carbon isotope ratio

Nana Suto[1], Hiroto Kawashima[2]

[1] Energy and Environment Research Division, Japan Automobile Research Institute, Tsukuba, 3050822, Japan
[2] Faculty of Systems Science and Technology, Akita Prefecture University, Yurihonjo, 0150055, Japan

*Correspondence to*: Nana Suto (nsuto@jari.or.jp) and Hiroto Kawashima (kawashima@akita-pu.ac.jp)

**Abstract.** The sources and seasonal trends of water-soluble organic carbon (WSOC) in carbonaceous aerosols are of significant interest. From July 2017 to July 2019, we collected samples of PM$_{2.5}$ (particulate matter, aerodynamic diameter < 2.5 μm)

from one suburban and one rural site in Japan. The average $\delta^{13}C_{WSOC}$ was −25.2 ± 1.1‰ and −24.6 ± 2.4‰ at the suburban site and rural site, respectively. At the suburban site, the $\delta^{13}C_{WSOC}$ was consistent with the $\delta^{13}C$ of burned C3 plants, and a high correlation was found between WSOC concentrations and non-sea-salt potassium concentrations; these results suggest that the main source of WSOC at this site was biomass burning of rice straw. At the rural site, the average $\delta^{13}C_{WSOC}$ was significantly heavier from autumn to spring (−23.9 ± 2.1‰) than in summer (−27.4 ± 0.7‰) ($p < 0.01$). The $\delta^{13}C_{WSOC}$ from autumn to spring

was consistent with that of biomass burning of rice straw, whereas that in summer was considered to reflect mainly the formation of secondary organic aerosols from biogenic VOCs. The heaviest $\delta^{13}C_{WSOC}$ (−21.3 ± 1.9‰) was observed from February to April 2019, which may be explained by long-range transport of C4 plant burning such as corn from overseas. Thus, the present study indicates that $\delta^{13}C_{WSOC}$ is potentially useful for elucidating the sources and atmospheric processes that contribute to seasonal variations of WSOC concentration.

**1 Introduction**

Particulate matter (PM) has deleterious effects on human health and contributes to climate change (Pope et al., 1995; Lohmann and Feichter, 2005). A major component of PM$_{2.5}$ (particulate matter, aerodynamic diameter < 2.5 μm) is carbonaceous aerosol, which comprises organic carbon (OC) and elemental carbon (EC) (Chow et al., 1993; Malm et al., 2004; Pöschl, 2005). The OC in carbonaceous aerosol can be further classified as water-insoluble organic carbon (WIOC) and water-

soluble organic carbon (WSOC) (Sullivan and Weber, 2006). WIOC is produced mainly by the combustion of fossil fuels and contains compounds such as alkanes (Pöschl, 2005). WSOC is emitted primarily from combustion processes, industrial process, and natural sources; it can also be formed through secondary processes such as homogeneous gas-phase or heterogeneous aerosol-phase oxidation (Claeys et al., 2004; Koch et al., 2007; Schichtel et al., 2008). WSOC accounts for 20%–80% of the total OC in carbonaceous aerosol depending on the location and season (Decesari et al., 2001; Sullivan et al., 2004; Du et al.,

2014; Duarte et al., 2015; Zhang et al., 2019). In addition, an average of 74% of all WSOC is contained in fine particles (Yu et al., 2004). WSOC is hygroscopic and therefore it enhances the capability of aerosols to act as cloud condensation nuclei, which affects climate change (Padró et al., 2010; Asa-Awuku et al., 2011). Therefore, source contributions of WSOC have been of significant interest for decades. A common approach for estimating the source contributions of WSOC is the use of a positive matrix factorization model. Using this approach, the annual contributions of biomass burning and secondary processes

to WSOC in Beijing, China, were estimated to be 40% and 54%, respectively (Du et al., 2014). Similarly, in Helsinki, Finland, the contribution of secondary organic aerosols (SOAs) to WSOC is reported to be high in summer (78%) but low in winter (28%) (Saarikoski et al., 2008). WSOC is known to contain various oxygenated compounds, including dicarboxylic acids, ketocarboxylic acids, aliphatic aldehydes, alcohols, saccharides, saccharide anhydrides, aromatic acids, phenols, amines, amino acids, organic nitrates, and organic sulfates (Duarte et al., 2007; Pietrogrande et al., 2013; Timonen et al., 2013; Chalbot

et al., 2014; Duarte et al., 2015). However, the precise molecular composition of WSOC is poorly understood because of the large number of compounds involved and the difficulties involved in identifying the individual components.

The stable carbon isotope ratio ($\delta^{13}$C) of carbonaceous aerosols can provide useful information about a sample of PM (e.g., Widory et al., 2004; Fisseha et al., 2009; Cao et al., 2011; Gensch et al., 2014). For example, because EC is unreactive, it is possible to identify its source directly from the $\delta^{13}$C of its aerosols (e.g., Kawashima and Haneishi, 2012; Zhao et al.,

2018). In contrast, because OC reacts in the atmosphere, its $\delta^{13}$C provides information not only about the source of the PM but also about any atmospheric processing it has undergone (e.g., Cao et al., 2011; Ni et al., 2018). In recent years, some groups have examined the $\delta^{13}$C of WSOC ($\delta^{13}C_{WSOC}$) in PM (Kirillova et al., 2010; Kirillova et al., 2013; Suto and Kawashima, 2018; Zhang et al., 2019). In addition, various approaches have been used; for example, the $\delta^{13}C_{WSOC}$ of ambient aerosols has been examined by means of wet oxidation with GasBench/isotope-ratio mass spectrometry (IRMS) (Fisseha et al., 2006) and by

means of combustion with an elemental analyzer/IRMS (EA/IRMS) (Kirillova et al., 2010). Recent advances have afforded highly sensitive analytical methods for determining $\delta^{13}C_{WSOC}$ values that use wet oxidation with liquid chromatography/IRMS (LC/IRMS) (Suto and Kawashima, 2018), GasBench/IRMS (Zhang et al., 2019), or total organic carbon analyzer/IRMS (Han et al., 2020); however, combustion-based approaches remain the most widely used.

The $\delta^{13}C_{WSOC}$ of particles of various sizes collected at various times of the year in East Asia (Miyazaki et al., 2012;

Kirillova et al., 2014a; Pavuluri and Kawamura, 2017; Yan et al., 2017; Suto and Kawashima, 2018; Zhang et al., 2019; Han et al., 2020), South Asia (Kirillova et al., 2013; Bosch et al., 2014; Kirillova et al., 2014b; Dasari et al., 2019), Europe (Fisseha et al., 2006; Fisseha et al., 2009; Kirillova et al., 2010), and the United States (Wozniak et al., 2012a; Wozniak et al., 2012b) have been reported (Table S1 in the Supplement). For example, the $\delta^{13}$C of total carbon ($\delta^{13}C_{TC}$) and $\delta^{13}C_{WSOC}$ of total suspended particles (TSP) was observed from September 2009 to October 2010 in Hokkaido, Japan (Pavuluri and Kawamura,

2017). Both $\delta^{13}C_{TC}$ and $\delta^{13}C_{WSOC}$ were heavier in winter than in summer, demonstrating seasonal variation. The authors concluded that the reason why $\delta^{13}C_{WSOC}$ was heavy in winter was because of the greater release of $^{13}$C by fossil fuel combustion and biomass burning. Similarly, Kirillova et al. (2013) collected TSP samples from January 2008 to April 2009 in Sinhagad, India, and Hanimaadhoo Island, Maldives. The average $\delta^{13}C_{WSOC}$ was $-20.4 \pm 0.5$‰ in Sinhagad and $-18.4 \pm 0.5$‰ in

Hanimaadhoo Island, which are heavier than values reported in other studies. In addition, aerosols reaching Hanimaadhoo Island after long-range, over-ocean transport were enriched by 3‰–4‰ in $\delta^{13}C_{WSOC}$ relative to the aerosols collected in Sinhagad. Based on these findings, Kirillova et al. (2013) reported for the first time that this enrichment of $\delta^{13}C$ was an effect related to the aging of OC during long-range transport of aerosol. Recent study reported that the enrichment of $\delta^{13}C_{WSOC}$ between source site (Delhi, India) and receptor site (Hanimaadhoo Island, Maldives) is caused by aging effect during long-range transport (Dasari et al., 2019).

The combustion method, which is widely used at present, requires more pretreatment time because samples of PM are extracted, dehydrated with a freeze drier, dried, and then measured by EA/IRMS. The wet oxidation/IRMS method described above do not require a drying stage during sample preparation; therefore, the total analysis time is markedly reduced compared with the combustion method. In addition, this newer approach is highly sensitive, so only small amounts of sample are needed compared to the combustion method. However, despite these improved approaches and the significant interest in the seasonal trends and source apportionment of WSOC, no studies have examined the change of $\delta^{13}C_{WSOC}$ in $PM_{2.5}$ over a long period of time to understand seasonal variability. As mentioned above, the small particle size $PM_{2.5}$ contains large number of WSOC, further investigations are needed. Here, we investigated the seasonal trends of WSOC at one suburban site and one rural site in Japan. Samples of $PM_{2.5}$ were collected from July 2017 to July 2019 at both sites, and $\delta^{13}C_{TC}$ and $\delta^{13}C_{WSOC}$ values, as well as carbon component and water-soluble ion concentrations, were determined. We then characterized the source of WSOC and any atmospheric processes it had undergone using isotope-based approaches. We believe that this is the first report of the use of the wet oxidation/IRMS method (Suto and Kawashima, 2018) for long-term observation of $\delta^{13}C_{WSOC}$.

## 2 Materials and experimental methods

### 2.1 Sampling sites and sample collection

Samples of $PM_{2.5}$ were collected at one suburban site and one rural site in Japan (Fig. S1 in the Supplement). The suburban site (Tsukuba, 36°4'N, 140°4'E) was on the rooftop of a 25-m-high building at the Japan Automobile Research Institute in Tsukuba City, Ibaraki Prefecture, Japan. Tsukuba is a suburban city located in the inland Kanto plain approximately 60 km northeast of the Tokyo metropolitan area. This site is surrounded by residential areas and forests, and there is a road in front of the building. $PM_{2.5}$ samples were collected approximately every 10 days from 19 July 2017 to 12 July 2019. The rural site (Yurihonjo, 39°23'N, 140°4'E) was on the campus of Akita Prefectural University in Yurihonjo City, Akita Prefecture, Japan. Yurihonjo is located 370 km northwest of Tsukuba and about 5 km away from the coast. The sampling site had no local pollutant sources such as large factories. Every year from December to February, the site is covered with several centimetres of snow (Japan Meteorological, 2019). $PM_{2.5}$ samples were collected approximately every 14 days from 11 August 2017 to 5 July 2019.

At both sites, the $PM_{2.5}$ samples were collected by using high-volume samplers (HV-1000F, Sibata Scientific Technology, Saitama, Japan) equipped with a $PM_{2.5}$ impactor (HV-1000-$PM_{2.5}$, Sibata Scientific Technology) at a flow rate of

approximately 1000 L min$^{-1}$. The samples were collected on quartz fiber filters (20.3 × 25.4 cm, 2500QAT-UP, Pallflex, Putnam, USA) that had been prebaked at 550 °C for 4 h before use. After sampling, the filters were kept in a freezer at −30 °C. A total of 107 PM$_{2.5}$ samples (62 samples from Tsukuba and 45 samples from Yurihonjo) were collected. PM$_{2.5}$ mass concentration was analyzed gravimetrically by using an electronic balance before and after sampling.

## 2.2 Stable carbon isotope ratio analysis

Determination of $\delta^{13}C_{TC}$ was performed at the Japan Automobile Research Institute using EA/IRMS (EA IsoLink, Thermo Fisher Scientific, Bremen, Germany; Delta V Advantage, Thermo Fisher Scientific, respectively). Portions of quartz filter (5–10 mg) were packed into a tin cup. The samples were combusted instantaneously with oxygen in the EA, and the carbon was converted to $CO_2$ via an oxidation/reduction tube of the EA. The oxidation/reduction tube and the packed column were maintained at 1020 °C and 60 °C, respectively. The flow rate of ultra-high-purity helium during the analysis was 180 mL min$^{-1}$. The $CO_2$ from the EA was ionized, and the $\delta^{13}C$ value was determined by means of IRMS; data acquisition was performed with Isodat software (ver. 3.0, Thermo Fisher Scientific).

Determination of $\delta^{13}C_{WSOC}$ was performed at Akita Prefectural University using the wet oxidation/IRMS method (Kawashima et al., 2018; Suto and Kawashima, 2018). A portion of each quartz fiber filter (14.13 cm$^2$) was extracted in 5 mL of Milli-Q water under ultrasonic agitation for 30 min. The extract was filtered through a syringe filter (Chromatodisc Type A 0.45 μm, GL Sciences, Japan) to remove insoluble material. The PM$_{2.5}$ samples were not decarbonated before $\delta^{13}C_{WSOC}$ analysis because the difference between the $\delta^{13}C_{WSOC}$ with and without hydrochloric acid pretreatment was within 0.2‰. A high-performance liquid chromatography (HPLC) system (Shimadzu Co.) was coupled to the IRMS instrument (Isoprime, Elementar UK, Manchester, UK) via a LiquiFace interface (Elementar UK). The HPLC system consisted of a column pump (LC-10ADvp), oxidation pump (LC-10ADvp), post-column pump (LC-10ADvp), autosampler (SIL-10ADvp), degasser (DGU-14A), and UV detector (SPD-10ADvp). The injection volume was 100 μL. The HPLC flow rate (without column), the sodium peroxodisulfate flow rate, and the post-column flow rate were 0.5, 0.4, and 0.3 mL min$^{-1}$, respectively. Sodium peroxodisulfate (0.5 M) and phosphoric acid (0.2 M) were mixed and then degassed in an ultrasonic bath for 1 h. One run took about 6 min. The trap current was set at 300 μA. The limits of detection (precision, <±0.3‰; accuracy, <±0.3‰) for levoglucosan and oxalic acid were 1111 and 1133 ngC, respectively.

The IRMS instrument and the data acquisition system were controlled by IonVantage NT software (ver. 1.5.4.0., Isoprime). The HPLC system was controlled by LCsolution software (ver. 1.25, Shimadzu Co.).

Stable carbon isotope ratios are expressed in δ notation in permil (‰)

$$\delta^{13}C\ [\text{‰}] = \left( \frac{R(^{13}C/^{12}C)_{sample}}{R(^{13}C/^{12}C)_{std}} - 1 \right) \qquad (1)$$

where $R(^{13}C/^{12}C)_{sample}$ and $R(^{13}C/^{12}C)_{std}$ (= 0.0111802) are the $^{13}C/^{12}C$ ratios for the sample and the standard (Vienna Pee Dee Belemnite), respectively. For all samples, the EA/IRMS and wet oxidation/IRMS data were measured in triplicate.

A two-point linear calibration was carried out for $\delta^{13}C$ (Coplen et al., 2006). For EA/IRMS, $\delta^{13}C_{TC}$ values and three internal laboratory standards were calculated by using the following international isotopic standards: IAEA-CH-3 (cellulose, $\delta^{13}C = -24.724‰$), IAEA-600 (caffeine, $\delta^{13}C = -27.771‰$), and USGS24 (graphite, $\delta^{13}C = -16.049‰$). These standards were obtained from the International Atomic Energy Agency (Vienna, Austria). As a check of instrumental stability, an isotope working standard (L-alanine, SI Science Co., Tokyo, Japan; $\delta^{13}C = -19.9‰$) was analyzed after every nine samples. For wet oxidation/IRMS, $\delta^{13}C$ values were calculated by means of a two-point linear calibration method from international isotope standards of sucrose (IAEA-CH-6, $\delta^{13}C = -10.449‰$), and three internal laboratory standards for D-(+)-arabitol ($\delta^{13}C = -23.6‰$), levoglucosan ($\delta^{13}C = -25.8‰$), and oxalic acid ($\delta^{13}C = -28.7‰$) obtained from EA/IRMS. Ultrapure water was prepared with a Milli-Q system (18.2 MΩ.cm; Millipore, Bedford, MA). To check instrumental stability, the laboratory standard of levoglucosan was analyzed after every nine samples. The average-1SD for $\delta^{13}C_{TC}$ and $\delta^{13}C_{WSOC}$ was 0.12‰ (<0.46‰) and 0.09‰ (<0.50‰), respectively, for all samples examined in the present study.

## 2.3 Chemical analysis

For determination of OC and EC concentrations, a portion of each quartz fiber filter (0.53 cm$^2$) was examined using a thermal-optical carbon analyzer (Model 2001, Desert Research Institute), and the samples were processed according to the IMPROVE Thermal Desorption/Optical Reflectance method with a 550 °C, split for OC and EC (Chow et al., 2001). The limits of detection for OC and EC were determined as three times the standard deviation of a blank filter, and they were 0.02 µg m$^{-3}$ and 0.02 µg m$^{-3}$, respectively. These limits of detection were sufficiently low (Yamagami et al., 2019). For determination of WSOC concentrations, a portion of each quartz fiber filter (1.58 cm$^2$) was extracted with 8 mL of ultrapure water for 30 min at room temperature. The water extracts were passed through a polyvinylidene difluoride filter (pore size 0.20 µm, GE Healthcare, USA) to remove insoluble materials, and then the filtrate was analyzed using a total organic carbon analyzer (TOC-L, Shimadzu, Kyoto, Japan). The limit of detection was determined as three times the standard deviation of a blank filter, and it was 0.03 µg m$^{-3}$, which was sufficiently low (Du et al., 2014). Quantification of the major water-soluble ions anions ($Cl^-$, $NO_2^-$, $NO_3^-$, $SO_4^{2-}$) and cations ($Na^+$, $NH_4^+$, $K^+$, $Mg^{2+}$, $Ca^{2+}$) was achieved by ion chromatography (Integrion RFIC; Thermo Fisher Scientific Inc., Sunnyvale, CA, USA). Details of water-soluble ion analysis method are described in Supplement S1.

## 3 Results and Discussion

### 3.1 Mass concentrations of PM$_{2.5}$ at the study sites

The average mass concentrations of PM$_{2.5}$ during the observation period were $19.7 \pm 8.2$ µg m$^{-3}$ (range, 7.1–46.6 µg m$^{-3}$) in Tsukuba and $11.2 \pm 4.7$ µg m$^{-3}$ (5.7–23.4 µg m$^{-3}$) in Yurihonjo (Table 1). The average mass concentration of PM$_{2.5}$ in Tsukuba was higher than the air quality standard for the annual average of Japan (15 µg m$^{-3}$) by the Ministry of the Environment and that at other residential sites across Japan (annual average in 2018, 11.2 µg m$^{-3}$) (Ministry of the Environment,

2019). In Yurihonjo, the average mass concentration of $PM_{2.5}$ was lower than the air quality standard for the annual average of Japan, and it was comparable with that at other residential sites across Japan.

A previous study reviewed the annual $PM_{2.5}$ concentrations in 45 global megacities in 2013 (Cheng et al., 2016). The five most-polluted megacities were Delhi, India; Cairo, Egypt; and Xi'an, Tianjin, and Chengdu, China ($PM_{2.5}$ annual average concentration, 89–143 µg m$^{-3}$). The five least-polluted megacities were Toronto, Canada; Miami, Philadelphia, and New York, United States; and Madrid, Spain ($PM_{2.5}$ annual average concentration, 7–10 µg m$^{-3}$). The mass concentration of $PM_{2.5}$ at both sites in the present study was much closer to that determined for the least-polluted megacities than that determined for the

most-polluted megacities. The mass concentrations of $PM_{2.5}$ in Tsukuba and Yurihonjo were significantly higher in winter and spring than in summer and autumn ($p < 0.01$). The mass concentrations of $PM_{2.5}$ were consistent with the seasonal variation for nearby sites of Atmospheric Environmental Regional Observation System (AEROS) provided by the Ministry of the Environment (Ministry of the Environment, 2021).

### 3.2 Concentrations of EC, OC, and WSOC, and OC/EC and WSOC/OC ratios

The concentrations of EC, OC, and WSOC, and the OC/EC and WSOC/OC ratios, at the study sites are summarized in Table 1. The concentrations of the carbon components (EC, WIOC, and WSOC) by season are shown in Fig. 1. The sum of EC and organic matter ($1.6 \times$ OC concentration) (Turpin and Lim, 2001) accounted for an average of 32% of the $PM_{2.5}$ mass concentration in Tsukuba and 25% in Yurihonjo. Thus, the contribution was slightly higher at Tsukuba than at Yurihonjo. The average EC concentration during the observation period was $0.9 \pm 0.4$ µg m$^{-3}$ (0.4–2.4 µg m$^{-3}$) in Tsukuba and $0.3 \pm 0.1$ µg

m$^{-3}$ (0.2–0.6 µg m$^{-3}$) in Yurihonjo. These values are comparable to those reported for Nagoya (1.1 µg m$^{-3}$) (Yamagami et al., 2019) and Niigata (0.5 µg m$^{-3}$) (Li et al., 2018), Japan, and lower than that reported for Xi'an, China (7.6 µg m$^{-3}$) (Zhao et al., 2018). The EC concentration contributed an average of 5% to the $PM_{2.5}$ mass concentration in Tsukuba and 3% in Yurihonjo. Currently, EC concentrations in Japan are decreasing as a result of Japanese government regulations on emissions from diesel vehicles (Yamagami et al., 2019). The average OC concentration during the observation period was $3.2 \pm 1.4$ µg m$^{-3}$ (1.0–6.6

180 µg m$^{-3}$) in Tsukuba and $1.5 \pm 0.8$ µg m$^{-3}$ (0.6–4.2 µg m$^{-3}$) in Yurihonjo. The OC concentration contributed an average of 28% to the $PM_{2.5}$ mass concentration in Tsukuba and 22% in Yurihonjo. The higher percentage contribution to the $PM_{2.5}$ mass concentration from OC than EC was in agreement with compared to other studies (Contribution of OC and EC concentration in $PM_{2.5}$ concentration: 20% and 6% in Korea) (Park and Cho, 2011).

      The OC/EC ratio is an indicator of the source of carbonaceous particles (Chow et al., 1996). The average OC/EC ratio

was $3.8 \pm 1.4$ in Tsukuba and $5.1 \pm 1.9$ in Yurihonjo. The higher OC/EC ratio at the rural site (Yurihonjo) than at the suburban site (Tsukuba) was comparable with the results of other studies (Ho et al., 2006; Zhang et al., 2008). This was likely because primary emissions, such as EC, are low at rural sites, meaning that the OC is larger in comparison. The high OC/EC ratio is due to the formation of secondary organic aerosols and biomass burning (Chow et al., 1996).

      The average WSOC concentration during the observation period was $1.2 \pm 0.4$ µg m$^{-3}$ (0.4–2.4 µg m$^{-3}$) in Tsukuba

and $0.8 \pm 0.5$ µg m$^{-3}$ (0.3–2.6 µg m$^{-3}$) in Yurihonjo. These values were similar to those reported for Sapporo (1.0 µg m$^{-3}$)

(Pavuluri and Kawamura, 2017) and Maebashi (2.3 µg m$^{-3}$) (Kumagai et al., 2009), Japan, but lower than those reported for Beijing, China (7.2 µg m$^{-3}$) (Du et al., 2014), and Gwangju, South Korea (3.7 µg m$^{-3}$) (Park and Cho, 2011). The WSOC concentration at Tsukuba was significantly higher in autumn and winter than in spring and summer ($p < 0.01$), whereas that in Yurihonjo was significantly higher in spring than in the other seasons ($p < 0.05$). The average WSOC/OC ratio was $0.4 \pm 0.1$ in Tsukuba and $0.5 \pm 0.1$ in Yurihonjo. This is consistent with previous studies that showed that the average WSOC/OC ratio was higher at rural sites than at urban sites (Kumagai et al., 2009; Ram and Sarin, 2010). This is also the same as the trend we found for OC/EC ratio in the present study.

### 3.3 $\delta^{13}C_{TC}$ and $\delta^{13}C_{WSOC}$

To our knowledge, this is the first report of a two-year-long observation of $\delta^{13}C_{TC}$ and $\delta^{13}C_{WSOC}$ in PM$_{2.5}$ at two sites simultaneously. $\delta^{13}C_{WSOC}$ values reported from previous studies conducted at various sampling sites and examining various particle sizes are summarized in Table S1. In the present study, the average $\delta^{13}C_{TC}$ was $-25.7 \pm 0.7$‰ ($-26.9$ to $-24.0$‰) in Tsukuba and $-24.7 \pm 1.6$‰ ($-27.3$ to $-20.4$‰) in Yurihonjo (Table 1 and Fig. 2). Previous studies have reported the average $\delta^{13}C_{TC}$ of TSP in Sapporo, Japan ($-24.8$‰ $\pm 0.68$‰) (Pavuluri and Kawamura, 2017), and of PM$_{2.5}$ in Sanjiang Plain, China ($-24.2$‰) (Cao et al., 2016), and these values are comparable to our present values.

In the present study, the average $\delta^{13}C_{WSOC}$ was $-25.2 \pm 1.1$‰ ($-26.7$ to $-21.8$‰) in Tsukuba and $-24.6 \pm 2.4$‰ ($-28.4$ to $-19.8$‰) in Yurihonjo (Table 1 and Fig. 2). The $\delta^{13}C_{WSOC}$ of PM$_{2.5}$, which was the particle size examined in the present study, was $-25.4$‰ $\pm 1.0$‰ in Delhi, India (Dasari et al., 2019), and $-24.2$‰ $\pm 0.6$‰ in Bhola, Bangladesh (Dasari et al., 2019), which are very close to our $\delta^{13}C_{WSOC}$ values. The $\delta^{13}C_{WSOC}$ of TSP was $-24.2$‰ $\pm 1.59$‰ in Sapporo, Japan (Pavuluri and Kawamura, 2017), $-24.0$‰ $\pm 1.5$‰ in Seoul, South Korea (Han et al., 2020), $-25.2$‰ $\pm 0.2$‰ in Millbrook, USA (Wozniak et al., 2012a), and similar values were obtained for particles of different sizes. In these previous studies, most of the average $\delta^{13}C_{WSOC}$ values were in the range of $-25$‰ to $-24$‰ regardless of particle size, although there were some heavy values such as those for Hanimaadhoo Island, Maldives ($-18.4$‰ $\pm 0.5$‰), and Sinhagad, India ($-20.4$‰ $\pm 0.5$‰) (Kirillova et al., 2013).

### 3.4 Seasonal variations in $\delta^{13}C_{TC}$ and $\delta^{13}C_{WSOC}$ in PM$_{2.5}$

$\delta^{13}C_{TC}$ and $\delta^{13}C_{WSOC}$ at Tsukuba showed no other clear seasonal variation, but they became slightly heavy from February to April 2019 (Fig. 2a). In contrast, the $\delta^{13}C_{TC}$ and $\delta^{13}C_{WSOC}$ at Yurihonjo were heaver from autumn to spring than in summer (Fig. 2b), and they showed a significant seasonal variation ($\delta^{13}C_{TC}$; $p < 0.01$, $\delta^{13}C_{WSOC}$; $p < 0.01$) compared to those in Tsukuba. In addition, $\delta^{13}C_{WSOC}$ became heavier from February to April 2019 as in Tsukuba. At both study sites, $\delta^{13}C_{WSOC}$ was usually heavier than $\delta^{13}C_{TC}$, but in summer $\delta^{13}C_{WSOC}$ was comparable to or lighter than $\delta^{13}C_{TC}$ (Tsukuba; $p < 0.01$, Yurihonjo; $p < 0.01$).

The seasonal trends of $\delta^{13}C_{TC}$ and $\delta^{13}C_{WSOC}$ observed in the present study were compared with those reported from previous long-term observations. No seasonal variation for $\delta^{13}C_{WSOC}$ in the suburban site, Tsukuba is comparable with that in

TSP in Seoul, South Korea, from March 2015 to January 2016 (Han et al., 2020). Similarly, clearly trend for heavier in winter than in summer for $\delta^{13}C_{TC}$ and $\delta^{13}C_{WSOC}$ in the rural site, Yurihonjo is comparable with that in TSP reported for Sapporo, Japan, from September 2009 to October 2010 (Pavuluri and Kawamura, 2017). In both Yurihonjo and Sapporo, it was observed that $\delta^{13}C_{WSOC}$ is usually heavier than $\delta^{13}C_{TC}$ and that this tendency is reversed in summer. Together, these findings imply that $\delta^{13}C_{WSOC}$ shows a weak seasonal trend in suburban or urban sites such as Tsukuba and Seoul, but a clear seasonal trend in rural sites such as Yurihonjo and Sapporo.

The variations (difference between maximum and minimum value) of $\delta^{13}C_{TC}$ and $\delta^{13}C_{WSOC}$ were 2.9‰ and 4.9‰ in Tsukuba and 7.0‰ and 8.6‰ in Yurihonjo, respectively. The variation of $\delta^{13}C_{WSOC}$ was larger than that of $\delta^{13}C_{TC}$ at both sites, with both variations larger in Yurihonjo. In previous studies, the variation of $\delta^{13}C_{TC}$ was reported as 2.5‰ in Sapporo (Pavuluri and Kawamura, 2017), and that of $\delta^{13}C_{WSOC}$ was reported as 5.5‰ in Sapporo (Pavuluri and Kawamura, 2017), and 6.5‰ in Seoul (Han et al., 2020). The variation of $\delta^{13}C_{EC}$ of $PM_{2.5}$ was only 1.6‰ in Japan (Kawashima and Haneishi, 2012) and 3.7‰ in China (Ni et al., 2018; Zhao et al., 2018). In the present study and these previous studies, the variation of $\delta^{13}C_{WSOC}$ was larger than that of $\delta^{13}C_{EC}$, regardless of sampling site. The reason for this is likely that $\delta^{13}C_{WSOC}$ is affected not only by the source characteristics but also by atmospheric processing. The reasons underlying the seasonal trend observed for $\delta^{13}C_{WSOC}$ are further discussed in Sections 3.5.1 and 3.5.2.

**3.5 Determination of seasonal trends and sources of WSOC using $\delta^{13}C_{WSOC}$**

**3.5.1 Seasonal trends and sources of WSOC in Tsukuba**

The average WSOC concentration in Tsukuba was significantly higher in autumn and winter than in spring and summer ($p < 0.01$), and EC concentrations showed a similar significant seasonal trend ($p < 0.01$) (Table 1). Table 2 shows the correlation coefficients between WSOC concentrations and three other parameters—$\delta^{13}C_{WSOC}$, EC concentration, and non-sea-salt potassium concentration (nss-$K^+$)—for each season and for the whole year. The EC concentration is a tracer of combustion (Bond et al., 2007). The nss-$K^+$ is a tracer of biomass burning that excludes $K^+$ from seawater (nss-$K^+$ = $[K^+]$ − 0.0335 × $[Na^+]$) (Lai et al., 2007). A weak correlation ($r = 0.18$) was found between the annual average WSOC concentration and annual average $\delta^{13}C_{WSOC}$. The strong correlation that was found between the annual average WSOC concentration and annual average EC concentration ($r = 0.71$) suggests that the WSOC at this suburban site is from combustion sources (e.g., fossil fuel and/or biomass burning). The strong correlations that were observed between WSOC concentrations and nss-$K^+$ for every season (autumn, $r = 0.96$; winter, $r = 0.83$; spring, $r = 0.85$; summer $r = 0.77$; all $p < 0.01$) further suggests that the WSOC at this site is a result of biomass burning. The dominant annual source for WSOC was consistent with that reported in Seoul by Han et al. (2020).

The average $\delta^{13}C_{WSOC}$ was −25.2 ± 1.1‰ in Tsukuba (Table 1). Because C3 and C4 plants have different metabolic pathways, their $\delta^{13}C$ values range from −34‰ to −24‰ for C3 plants and from −19‰ to −6‰ for C4 plants (Smith and Epstein, 1971). When C3 plants are burned in the laboratory, there is no difference between the $\delta^{13}C$ of the produced particles and that

of the original C3 plants (Turekian et al., 1998; Das et al., 2010). However, the particles produced by burning C4 plants are 3.5‰ lighter than the original C4 plants (Turekian et al., 1998). Therefore, the $\delta^{13}C$ of C4 plants was estimated to be −22.5 to −9.5‰. The $\delta^{13}C$ of C3 and C4 plant burning has been estimated to be −34.7 to −25.1‰ and −19.3 to −16.1‰, respectively (Kawashima and Haneishi, 2012; Garbaras et al., 2015; Guo et al., 2016). Thus, the average $\delta^{13}C_{WSOC}$ at Tsukuba indicates that the burning of C3 plant biomass is the dominant source of WSOC at this site. Indeed, rice, a C3 plant, is Japan's largest crop followed by barley and wheat (Ministry of Agriculture Forestry and Fisheries, 2018). In Ibaraki Prefecture, where Tsukuba City is located, the crop acreage and harvest of rice was 68,400 ha and 358,400 tons in 2018 were the largest in the Kanto Region (Ministry of Agriculture Forestry and Fisheries, 2018). In addition, according to a field investigation, biomass burning in Tsukuba is predominantly the burning of rice straw and rice hulls from September to October (Tomiyama et al. (2017). Using radiocarbon analysis, which can distinguish between biogenic and anthropogenic sources, a higher proportion of OC in $PM_{2.5}$ collected in Tokyo, Japan, in 2014 was reported to be biogenic from autumn to winter than in summer (Hoshi and Saito, 2020).

The main chemical component generated by the breakdown of cellulose by biomass burning is levoglucosan, which can be used as a tracer of biomass burning (Simoneit et al., 1999). The $\delta^{13}C$ of levoglucosan emitted from the burning of C3 plants such as peanut, mulberry, China fir, Chinese red pine, Chinese guger tree, and Chestnut are reported to range from −26.05 to −22.60‰, with that from rice straw reported to be −24.26 ± 0.09‰ (Sang et al., 2012). The average $\delta^{13}C_{WSOC}$ in Tsukuba was very close to this previously reported $\delta^{13}C$ of levoglucosan from the burning of rice straw. However, levoglucosan accounts for only about 3.8% of the WSOC in urban areas of Japan (Kumagai et al., 2010). Therefore, it is difficult to accurately identify the sources of WSOC using only the $\delta^{13}C$ values of levoglucosan. Further research is needed to determine the $\delta^{13}C$ of the components of WSOC other than levoglucosan.

### 3.5.2 Seasonal trends and sources of WSOC in Yurihonjo

In Yurihonjo, the correlation between WSOC concentrations and EC concentrations was highest in winter ($r = 0.87$, $p < 0.01$), followed by autumn ($r = 0.83$, $p < 0.01$) and spring ($r = 0.64$, $p < 0.05$), and lowest in summer ($r = 0.24$) (Table 2). This suggests that WSOC at this rural site was mainly from combustion sources in autumn and spring. In addition, the correlation between WSOC concentrations and nss-$K^+$ concentrations and was very high in autumn ($r = 0.93$), winter ($r = 0.99$), and spring ($r = 0.80$; all $p < 0.01$) but not in summer ($r = 0.40$). These strong correlations from autumn to spring suggest that during that time the WSOC came mainly from combustion sources such as biomass burning. The average $\delta^{13}C_{WSOC}$ at Yurihonjo for autumn and spring, −23.9 ± 2.1‰, suggests that biomass burning of C3 biomass such as rice straw and rice hulls may be the dominant source of WSOC, as was found in Tsukuba.

In Akita Prefecture, where Yurihonjo is located, the crop acreage of rice was 87,700 ha in 2018, and the rice harvest was 491,100 tons (Ministry of Agriculture Forestry and Fisheries, 2018). From February to April 2019, the $\delta^{13}C_{WSOC}$ was the heaviest (−21.3 ± 1.9‰), and WSOC concentrations were markedly increased compared with the previous months (average, 1.5 ± 0.7 μg m$^{-3}$) (Fig. 1b and Fig. 2b). A moderate correlation between WSOC concentrations and $\delta^{13}C_{WSOC}$ values was

observed for this time period ($r = 0.54$, $p = 0.27$). This $\delta^{13}C_{WSOC}$ value indicates a heavy $\delta^{13}C$ source such as C4 plants (e.g., corn and grass), but no evidence of burning of C4 plants during this period was observed around the sampling site at Yurihonjo.

Northeast China is the largest producer of corn in China (MWCACP, 2019), and biomass burning is used for heating in winter (Chen et al., 2017). Satellite imagery revealed a number of fire spots in that part of China from February to April 2019 (NASA, 2017) (Figure S2 in the Supplement). Backward air-mass trajectories showed that air masses at Yurihonjo during this period originated mainly from areas in northeast China (Fig. S3 in the Supplement). Consistent with this finding, Uranishi et al. (2020) reported from an analysis using the Community Multiscale Air Quality model that particles from biomass burning in northeast

China were transported to Akita Prefecture in February and March 2019. The correlation between $Na^+$ and $Cl^-$ concentrations was highest from winter to spring 2019 in Yurihonjo ($r = 0.98$, $p < 0.01$), suggesting the influence of sea salt. Recently, aerosol photochemical aging during long-range transport has been shown to selectively enrich the $^{13}C$ content in organic aerosols, leading to heavier $\delta^{13}C$ values (Kirillova et al., 2013; Bosch et al., 2014; Dasari et al., 2019; Zhang et al., 2019). In a field study, the isotope fractionation values for $\delta^{13}C_{WSOC}$ were estimated to be enriched by 3‰–4‰ because of aging during

transport (Kirillova et al., 2013). The combination of isotopic ratio and concentration measurements (Fig. 1 and Fig. 2) together with the evidence of prevailing biomass burning activities (Fig. S2) and the results of the backward trajectory analysis (Fig. S3) suggest that the heavier $\delta^{13}C_{WSOC}$ from February to April 2019 at Yurihonjo was the result of C4 plant combustion rather than aging during long-range transport.

The $\delta^{13}C_{WSOC}$ in summer was very light ($-27.4‰$) compared with the average value for the observation period. A

weak correlation between WSOC concentrations and EC concentrations in summer ($r = 0.24$; Table 2) suggests that the WSOC is derived from a non-combustion source. In general, the formation of WSOC involves atmospheric reactions such as the formation of SOAs, which are formed by oxidation of anthropogenic and biogenic VOCs (Heo et al., 2013). Aliphatic hydrocarbons (e.g., alkanes and alkenes) and aromatics (e.g., benzene, toluene, ethylbenzene, and xylene) emitted from solvent evaporation and vehicle emissions are important anthropogenic VOCs and precursors of SOAs (Chen et al., 2010). The $\delta^{13}C$

values for alkanes in tunnel, gas station, underground garage, and refinery air samples are reported to range from $-28.6 \pm 1.8‰$ to $-27.3 \pm 2.1‰$ (Rudolph et al., 2002). Toluene and xylene are the aliphatic hydrocarbons with the highest annual emissions in Japan (Japan Ministry of Economy Trade and Industry, 2020). The $\delta^{13}C$ of toluene and xylene for gas station and vehicle emissions are reported to range from $-27.7‰$ to $-23.8‰$ (Rudolph et al., 2002; Kawashima and Murakami, 2014). Because VOCs in the atmosphere are oxidized by photochemical oxidants, the $\delta^{13}C$ values of the residual VOCs become heavier via

isotopic fractionation (Rudolph et al., 2000; Anderson et al., 2004); in other words, secondary production tends to result in a lighter $\delta^{13}C$ for SOA in the atmosphere. In a laboratory-based experiment, the $\delta^{13}C$ of SOA particles formed by photooxidation of toluene was 3‰ to 6‰ lighter than that of the precursor toluene, depending on the degree of oxidation (Irei et al., 2006; Irei et al., 2011). Assuming that this isotopic fractionation of toluene applies also to all other potential components, the $\delta^{13}C$ of the emission source of VOCs at Yurihonjo would be approximately $-24.4$ to $-21.4‰$, as calculated by subtracting 3‰ to 6‰

from the average $\delta^{13}C_{WSOC}$ in Yurihonjo during summer ($-27.4‰$). This estimated $\delta^{13}C$ value of VOCs is heavier than those

previously reported for anthropogenic VOCs. Therefore, anthropogenic VOCs were not considered to be the dominant source of WSOC at Yurihonjo.

At the global scale, biogenic VOC emissions are more than an order of magnitude greater than those of anthropogenic VOCs (Farina et al., 2010). Biogenic VOCs include isoprene, monoterpenes, and sesquiterpenes released from vegetation, with isoprene producing the most SOA (Atkinson and Arey, 1998). The oxidation product of isoprene is 2-methyltetrol, which is widely used as an organic tracer to evaluate the production of SOA from isoprene (Claeys et al., 2004). The average $\delta^{13}$C of 2-methyltetrol in aerosols in four forests in Sichuan Province, China, was $-27.36‰$ ($-28.23$ to $-26.46‰$) (Li et al., 2019). This average $\delta^{13}$C of 2-methyltetrol is close to the $\delta^{13}C_{WSOC}$ detected in summer in Yurihonjo, suggesting the components produced by secondary reaction of biogenic VOCs make a large contribution to the WSOC in Yurihonjo during the summer. From a field study conducted in a forest in northern Japan, Miyazaki et al. (2012) reported that the lightest $\delta^{13}C_{WSOC}$ values (average $-25.6 \pm 0.7‰$) were observed from June to September; the authors concluded from positive matrix factorization modelling data that biogenic SOAs (isoprene SOA and $\alpha$-/$\beta$-pinene) were the dominant source of WSOC in the summer, which is consistent with the findings of the present study.

**4 Conclusion**

The WSOC concentration, $\delta^{13}C_{TC}$, and $\delta^{13}C_{WSOC}$ of PM$_{2.5}$ were observed at one suburban and one rural site in Japan over a two-year period. The average WSOC concentration during the observation period was $1.2 \pm 0.4$ µg m$^{-3}$ ($0.4$–$2.4$ µg m$^{-3}$) at the suburban site and $0.8 \pm 0.5$ µg m$^{-3}$ ($0.3$–$2.6$ µg m$^{-3}$) at the rural site. The $\delta^{13}C_{WSOC}$ was $-25.2 \pm 1.1‰$ ($-26.7$ to $-21.8‰$) at the suburban site and $-24.6 \pm 2.4‰$ ($-28.4$ to $-19.8‰$) at the rural site. The $\delta^{13}C_{TC}$ and $\delta^{13}C_{WSOC}$ at the suburban site showed no clear seasonal variations, but they were slightly heavier from February to April 2019. In contrast, the $\delta^{13}C_{TC}$ and $\delta^{13}C_{WSOC}$ at the rural site were heaver from autumn to spring than in summer, and they showed a significant seasonal variation ($\delta^{13}C_{TC}$, $p < 0.01$; $\delta^{13}C_{WSOC}$, $p < 0.01$). Using $\delta^{13}C_{WSOC}$, carbon components, and water-soluble ions, the main source of WSOC at the suburban site was concluded to be local biomass burning of rice straw. At the rural site, the $\delta^{13}C_{WSOC}$ from autumn to spring was concluded to reflect mainly the biomass burning of rice straw, whereas that in summer was considered to reflect mainly the formation of secondary organic aerosols from biogenic VOCs. The heaviest $\delta^{13}C_{WSOC}$ ($-21.3 \pm 1.9‰$) was from February to April 2019 and may reflect long-range transport of particles resulting from the overseas burning of C4 plants such as corn. Thus, we were able to use a $\delta^{13}C_{WSOC}$-based approach to understand the sources and atmospheric processes that contribute to the WSOC concentrations at the two study sites.

*Data availability.* Data are available from the corresponding author on request (nsuto@jari.or.jp).

*Author contribution.* NS and HK were involved in research planning and experimental design. NS performed the sampling and measurements of $\delta^{13}C_{TC}$, carbon components and water-soluble ions. HK performed the sampling and measurements of $\delta^{13}C_{wsoc}$. All authors clarified the experimental data and contributed to the writing of the paper.

*Competing interests.* The authors declare that they have no conflict of interest.

*Acknowledgements.* This work was partially supported by the Japan Society for the Promotion of Science KAKENHI Grant Numbers 19K20463, 18H03393, 17K12829, and 16KK0015. We acknowledge the use of data and imagery from NASA's Fire Information for Resource Management System (FIRMS) (https://earthdata.nasa.gov/firms), part of NASA's Earth Observing System Data and Information System (EOSDIS). We thank emeritus professor Shigeki Masunaga of Yokohama National University for providing the high-volume samplers used in this research, and Sae Ono, Momoka Suto, and Otoha Yoshida for collecting the aerosol samples and for helping wet oxidation/IRMS analysis at Akita Prefectural University. Furthermore, Dr. Akiyoshi Ito, Dr. Hiroyuki Hagino, Kazue Kagami and Akemi Nakayama at Japan Automobile Research Institute, for their advice and help with chemical analysis. And, Yumi Sone from Thermo Fisher Scientific Inc., Japan was very helpful with our EA/IRMS analysis. Finally, we thank from ELSS, Inc. for editing the English of this manuscript.

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

**(a) Tsukuba, Ibaraki**

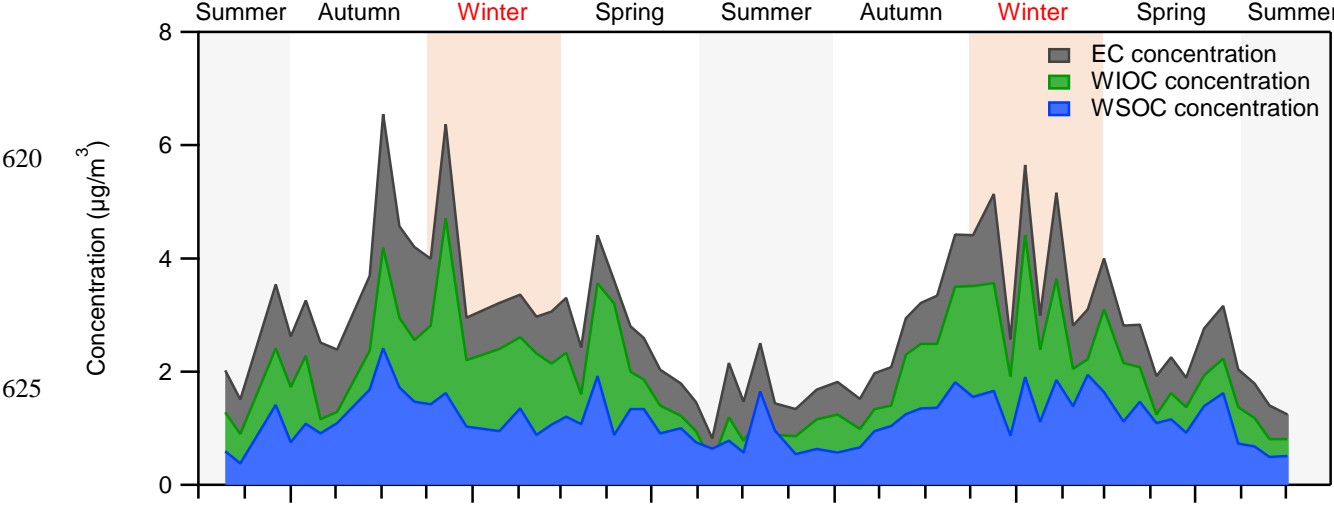

**(b) Yurihonjo, Akita**

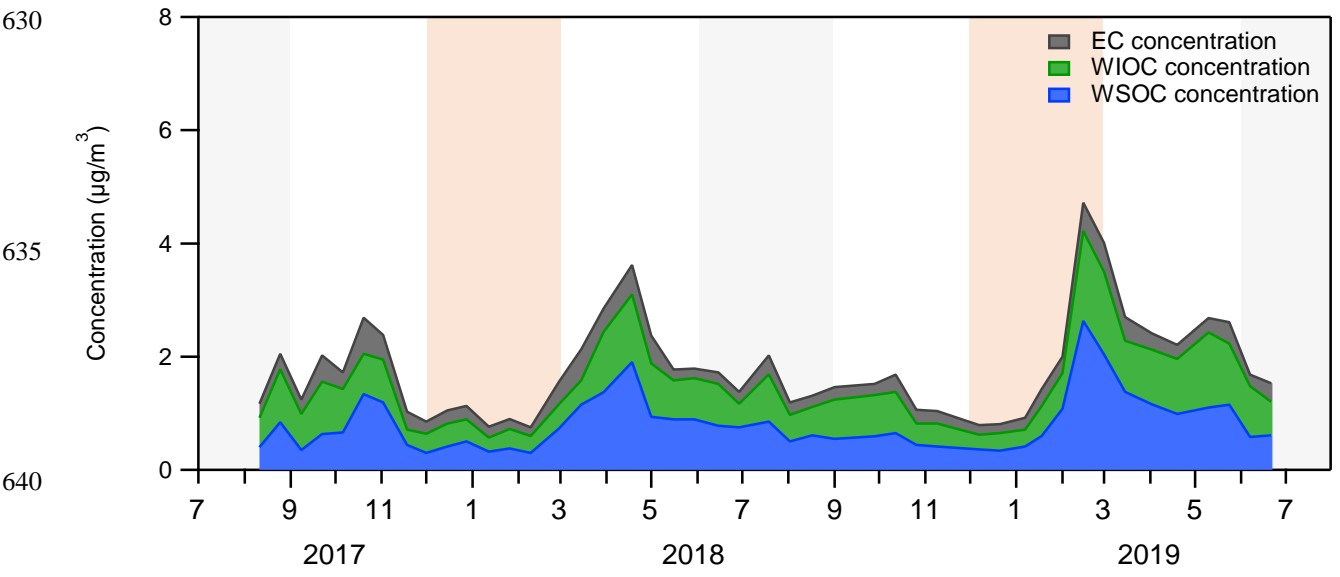

Figure 1: Concentrations of EC, WIOC, and WSOC of PM$_{2.5}$ from July 2017 to July 2019 in (a) Tsukuba, Ibaraki, and (b) Yurihonjo, Akita, Japan.

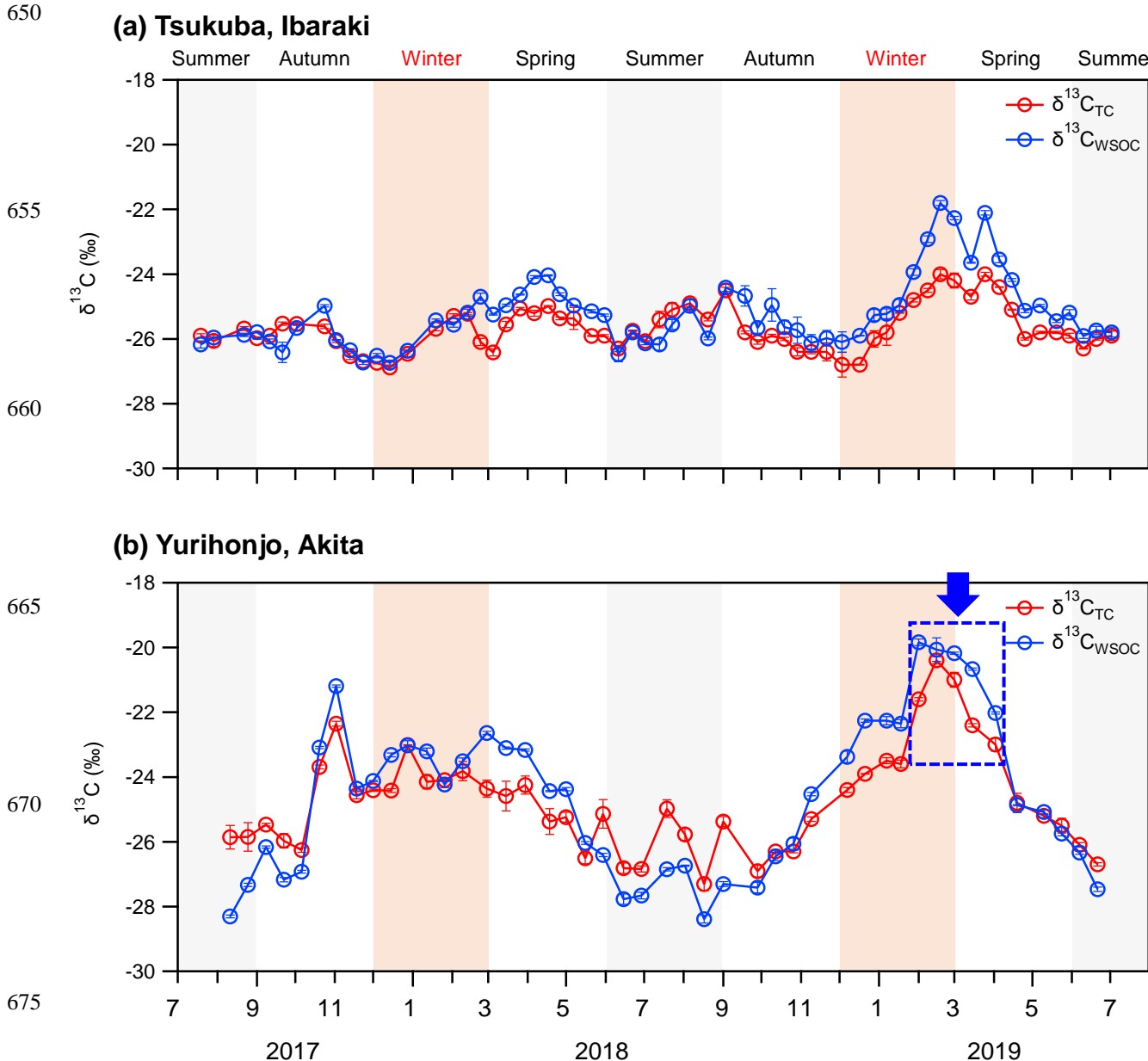

**Figure 2: δ¹³C<sub>TC</sub> and δ¹³C<sub>WSOC</sub> of PM₂.₅ from July 2017 to July 2019 in (a) Tsukuba, Ibaraki, and (b) Yurihonjo, Akita, Japan.**

**Table 1.** Seasonal average concentrations of $PM_{2.5}$, EC, OC, and WSOC; OC/EC and WSOC/OC ratios; and $\delta^{13}C_{TC}$ and $\delta^{13}C_{WSOC}$ values for $PM_{2.5}$, in Tsukuba and Yurihonjo, Japan.

Tsukuba

| Compound | Season (average ± SD) | | | | Average ($n = 62$) |
|---|---|---|---|---|---|
| | Spring ($n = 18$) | Summer ($n = 13$) | Autumn ($n = 16$) | Winter ($n = 15$) | |
| $PM_{2.5}$ (µg m$^{-3}$) | 23.5 ± 7.7 | 14.4 ± 4.1 | 16.5 ± 7.2 | 23.0 ± 8.9 | 19.7 ± 8.2 |
| EC (µg m$^{-3}$) | 0.7 ± 0.2 | 0.7 ± 0.3 | 1.1 ± 0.5 | 1.0 ± 0.4 | 0.9 ± 0.4 |
| OC (µg m$^{-3}$) | 3.2 ± 1.0 | 1.8 ± 0.8 | 3.4 ± 1.4 | 4.2 ± 1.2 | 3.2 ± 1.4 |
| WSOC (µg m$^{-3}$) | 1.2 ± 0.3 | 0.8 ± 0.4 | 1.3 ± 0.5 | 1.4 ± 0.4 | 1.2 ± 0.4 |
| OC/EC | 4.5 ± 1.6 | 2.7 ± 0.6 | 3.5 ± 1.2 | 4.4 ± 0.8 | 3.8 ± 1.4 |
| WSOC/OC | 0.4 ± 0.1 | 0.4 ± 0.1 | 0.4 ± 0.0 | 0.3 ± 0.1 | 0.4 ± 0.1 |
| $\delta^{13}C_{TC}$ (‰) | −25.3 ± 0.7 | −25.8 ± 0.4 | −26.0 ± 0.5 | −25.7 ± 0.9 | −25.7 ± 0.7 |
| $\delta^{13}C_{WSOC}$ (‰) | −24.4 ± 1.0 | −25.9 ± 0.4 | −25.7 ± 0.6 | −25.1 ± 1.4 | −25.2 ± 1.1 |

Yurihonjo

| Compound | Season (average ± SD) | | | | Average ($n = 45$) |
|---|---|---|---|---|---|
| | Spring ($n = 12$) | Summer ($n = 9$) | Autumn ($n = 11$) | Winter ($n = 13$) | |
| $PM_{2.5}$ (µg m$^{-3}$) | 15.8 ± 4.2 | 8.6 ± 2.4 | 8.1 ± 1.2 | 11.4 ± 5.1 | 11.2 ± 4.7 |
| EC (µg m$^{-3}$) | 0.4 ± 0.1 | 0.2 ± 0.1 | 0.3 ± 0.1 | 0.2 ± 0.1 | 0.3 ± 0.1 |
| OC (µg m$^{-3}$) | 2.2 ± 0.6 | 1.3 ± 0.3 | 1.3 ± 0.5 | 1.1 ± 1.0 | 1.5 ± 0.8 |
| WSOC (µg m$^{-3}$) | 1.2 ± 0.4 | 0.7 ± 0.2 | 0.7 ± 0.3 | 0.6 ± 0.6 | 0.8 ± 0.5 |
| OC/EC | 6.6 ± 2.1 | 5.5 ± 1.5 | 4.2 ± 1.2 | 4.1 ± 1.5 | 5.1 ± 1.9 |
| WSOC/OC | 0.6 ± 0.1 | 0.5 ± 0.1 | 0.5 ± 0.1 | 0.6 ± 0.1 | 0.5 ± 0.1 |
| $\delta^{13}C_{TC}$ (‰) | −24.4 ± 1.6 | −26.2 ± 0.7 | −25.3 ± 1.3 | −23.5 ± 1.2 | −24.7 ± 1.6 |
| $\delta^{13}C_{WSOC}$ (‰) | −23.8 ± 2.0 | −27.4 ± 0.7 | −25.5 ± 2.0 | −22.6 ± 1.3 | −24.6 ± 2.4 |

**Table 2.** Correlation ($r$) between WSOC concentration and the stated parameters.

| Season | Tsukuba | | | Yurihonjo | | |
|---|---|---|---|---|---|---|
| | $\delta^{13}C_{WSOC}$ | EC | nss-K$^+$ | $\delta^{13}C_{WSOC}$ | EC | nss-K$^+$ |
| Spring | 0.36 | 0.73 | 0.85 | 0.63 | 0.64 | 0.80 |
| Summer | −0.14 | 0.84 | 0.77 | 0.17 | 0.24 | 0.40 |
| Autumn | −0.45 | 0.75 | 0.96 | 0.65 | 0.83 | 0.93 |
| Winter | 0.29 | 0.68 | 0.83 | 0.77 | 0.87 | 0.99 |
| Annual | 0.18 | 0.71 | 0.88 | 0.44 | 0.72 | 0.87 |