# Peer review of "Measurement report: Source characteristics of water-soluble organic carbon in PM2.5 at two sites in Japan, as assessed by long-term observation and stable carbon isotope ratio"

_Atmospheric Chemistry and Physics, 2020_

## Referee Comment (RC1) · Anonymous Referee #1 · 10 Dec 2020

This paper presents long-term concentration and isotopic ratio measurements of TC and WSOC in ambient PM2.5 collected at two sites in Japan between July 2017 and July 2019. The authors show a quite impressive series of measurements aiming to investigate sources of the aerosol fine fraction at a suburban and a rural background site. Using stable isotope analyses represents a novel approach for source characterization, thus, is suitable for this goal. The wet oxidation prior to IRMS is very challenging but opens up the opportunity to much easier separate polar compounds and measure their isotopic ratios.

[Figure]

Unfortunately, the presentation is on a poor level. Therefore, it needs to be substantially improved before publishing.

General comments:

Far too little emphasis is placed on the contribution of the isotope measurements to elucidate aerosol sources or chemical processing during atmospheric transport. The main criticism for this manuscript is that the isotopic discussions are generally kept at a very superficial level. The lines of reasoning are often vague, sometime contrived.

Specific comments:

1) The authors present d13C for WSOC and TC. They compare their observations with isotopic ratios of single tracers such as for levoglucosan or toluene, which is quite senseless. As for the former (Sections 3.5.1 and 3.5.2 as well as in the abstract), even during intense biomass burning activities, levoglucosan contributes with few percent to OC and consequently, with less then 10% to the WSOC. Due to the complexity of the biomass burning sources and processes, it is very unlikely that the levoglucosan source specific d13C (please note here that Sang et al. EST2012 presented d13C0 of levoglucosan in aerosol formed during the combustion of C3 plants only) will determine alone the WSOC or TC d13C. The arguments used to interpret the isotopically lighter aerosol in summer are laboured, too. The authors cite here Irei et al., who investigated isotopic ratios of different generation reaction products of toluene oxidation. The depletion by 6 permille is valid only for the mentioned reaction (since KIE epsilon is 5.95 permille) and only for its early stages. For other fossil combustion tracers, oxidation reactions show a large range of KIE (Anderson et al. GRL2004), therefore their products will be very differently depleted at the reaction beginning. Compound specific isotope measurements of single tracers are necessary for detailed studies. Yet, the TC and WSOC d13C can be compared with corresponding values (Gensch et al. IJMS2014). The observations agree mainly with TC and WSOC d13C of C3 plant and fossil fuel combustion aerosol.

2) As for the TC and WSOC d13C seasonal trend, the authors state in Lines220-224: 'd13CTC and d13CWSOC at Tsukuba became slightly heavy from February to April 2019, but they showed no other clear seasonal variation (Fig. 2a). In contrast, the d13CTC and d13CWSOC at Yurihonjo were heaver in winter and spring than in summer and autumn (Fig. 2b), and they showed a significant seasonal variation (d13CTC; p < 0.01, d13CWSOC; p < 0.01) compared to those in Tsukuba. At both study sites, d13CWSOC was usually heavier than d13CTC, but in summer d13CWSOC was comparable to or lighter than d13CTC.' Firstly, most of the differences between TC and WSOC d13C seem to be within the uncertainty range for isotopic measurements. To show the opposite, the authors should present some statistical evidence. Secondly, the unlike seasonal variation between the two sites give some additional information, which should be discussed in more detail. In absence of compound specific analyses, there are only qualitative indications, but exactly such discussions (e.g. local sources close by the suburban sampling site, which 'flatten' the influence of the long range transport) would enlighten the advantages of using stable isotopes in atmospheric studies. The enrichment by up to 6 permille in winter cannot be explained by chemistry alone, considering the lower oxidant concentration in the cold season. Rather, it looks like significant contribution of heavier sources to the collected aerosol (coal or even C3 plant combustion, see Gensch et al. IJMS2014). This assumption is supported by the similar d13C for TC and WSOC. A back-trajectory analysis would help to elucidate such questions.

3) Line124: remove 'units, were calculated as follows'. Suggestion: 'Stable carbon isotope ratios are expressed in terms of $\delta$ notation in permil (‰:' Equation 1 gives only the meaning of d13C. In the lab, CO2 working standards are used (calibrated against IAEA standards). The final d13C values are reported relatively to the international reference VPDB. The calculations behind are described in Brand et al. PureApplChem2010.

Editorial revisions:

The authors should consider renouncing to mention the used computer OS (Lines107,

122, 123)

Generally, the used English is not optimal. I strongly suggest that this manuscript is carefully revised by a native speaker.

Some examples of the inadequate language: 1) Unhandy expressions - Lines38-39: 'Although it is possible to estimate the contribution rate using PMF, it is necessary to identify the characterisation of source artificially.' What is the meaning of 'artificially'? - Line91: ' Every December to February' 2) Confusing indications: - Lines157, 159: '... than the Japan Environmental Standard for the annual average...' It is surely meant a threshold stipulated in the air quality guidelines of the Japan Environmental...organisation. 3) Wrong wording - Replace 'reasonable' describing the measurements (lines171, 186). Use instead 'good agreement', 'similar to other studies', 'as expected'... - Lines226-227: Rephrase: 'd13CWSOC in TSP in Seoul, South Korea, from March 2015 to January 2016 showed no seasonal variation (Han et al., 2020), which is comparable with our present findings for the suburban site, Tsukuba.' The findings of this work should be presented first and then compare them with other studies.

---

## Referee Comment (RC2) · Andrius Garbaras (Referee) · 2 Jan 2021

The manuscript entitled "Measurement report: Source characteristics of water-soluble organic carbon in PM2.5 at two sites in Japan, as assessed by long-term observation and stable carbon isotope ratio" by N. Suto and H. Kawashima represents investigation of aerosol particles through the prism of carbon stable isotopes, EC, OC and ion analysis. The sample collection was performed during two years, which allowed for the authors to catch the seasonal changes in some aerosol parameters.

[Figure]

In general, the manuscript is a bit chaotic, sometimes pointing at not relevant issues. Also, I think Authors have not exploited the benefits of long time samples collection and analysis.

I would like to address the issues that I am concerned on:

Lines 25-30 In the introduction part, the most recent reference is from the year 2011.

Line 35. Why authors mention about PMF? They are not using PMF in this manuscript? What it means "it is necessary to identify the characterisation of source artificially"?

Line 45. In general, the isotope ratio literature review part is too narrow in this manuscript. I propose to make overview on most recent references, which deals with aerosols and stable isotopes in different areas, which are relevant for this manuscript: transportation, natural terrestrial and marine (also oceanic) sources, fires etc. Suggestions: Zhang W. et al., 2019, ACP 19, 11071-11087 ($\delta$13C of WSOC in China); Fisseha R. et al 2006, Atmos. Env. 43 431-437 ($\delta$13C values in the different aerosol fractions in Zurich, Switzerland); Kirillova E.N. et al. 2010 Anal. Chem. 82 7973-7978; Lang S.Q. et al. 2012 Rapid Commun Mass Sp. 26 9-16; Suto N. and Kawashima H. Rapid Commun Mass Sp. 32 1668-1674 (In this manuscript the same Authors present the own developed method for $\delta$13C in WSOC). The last citation is used later, but I propose to cite it here also, as it is relevant to this part.

Section 2.1. I see that present study two sampling points were separated by a distance of 370 km . Both Tsukuba and Yurihonjo has a Humid continental climate, with different annual mean temperature of about 2°C. Tsukuba is close to Ocean, while Yurihonjo is close to Sea of Japan. I assume that air mass back trajectories are important when comparing two sampling points in this case. I assume that can be problematic to compare suburban or rural places, as they can be influenced by totally different atmosphere (and aerosol sources) conditions. In the rest of the text I'm missing marine source evaluation (except nss-K+).

Line 95. I did not understand how long sampling for one filter was performed, does all filters were loaded periodically? Maybe worth to add as supplementary filter data (time, weight of loaded aerosol). How Authors ensured that the filter not lost mass itself, as quartz filters break easily, and making gravimetric measurements can be complicated. Does Authors used equilibration for the unexposed and exposed filter to account for humidity?

Section 2.2 can be improved. I miss a clear step-by-step description of analytical procedures: Line 105: "...the carbon was converted to CO2 via an oxidation catalyst in the reduction tube of EA" - I think is incorrect and must be clarified. Does oxygen was added? Does nitrogen was seperated from CO2? What about blank measurements, does it gave any signal? In this section, is no clear separation, which part of the section corresponds to total carbon measurements, which part to WSOC?

Line 170. Authors state that snow is responsible for low PM2.5 concentration in Yurihonjo. My question, does land in winter emits significant amount of aerosol, even not covered by snow? What about heating activities, long range transport etc. I want to say that Authors must to think generally what they want to deliver to the reader, and maybe skip parts which are not relevant to the main purpose of the article (for example about most polluted megacities, as here megacities are not investigated). Yurihonjo can't be compared to a megacity, as by definition it is an urban site (last line in 165).

Line 260-265. The assumption that rice residues burning is the main source of WSOC in Tsukuma is nice assumption, but I was not convinced by this statement. Usually, in urban sites weak seasonal trend in carbon delta values is due to transportation influence. Here, Authors did not evaluate the $\delta 13C$ of fossil fuel emissions. In addition, assuming that crop combustion is the main source of WSOC, we must to see correlation with delta values. No correlation between WSOC concentrations and delta values means that does not exist one dominating aerosol source. In other words, if in Tsukuma main source would be crop burning, we must observe correlation between WSOC and delta values. Or, delta values are affected by atmospheric processing, but

then air mass back trajectory analysis (even with Solar intensity) must be performed to understand the resident time of the aerosol.

Line 290. Speculations about long-range transport must be verified by air mass trajectory analysis. Also, Authors can use other delta signatures (various crops, land and forest fires, transportation and fuel burning emissions, shipping emissions etc.), not only rise burning.

Line 310. Kawashima and Murakami 2014 report that in roadside samples VOC ranged from -29.6 to 23.5 permill (the same range for ambient samples). It makes difficulty interpreting vehicular emissions from a stable isotopes perspective. I'm not sure if only Toluene and Xylene are responsible for WSOC in Yurihonjo, so 5.8 permill subtraction must be made carefully (Authors note themselves that anthropogenic emissions of VOC are smaller than that biogenic).

Figure 2. I not found in the manuscript explanation, what causes enriched delta values in the winter in season 2018/2019 compared to 2017/2018 at both locations?

---

## Author Comment (AC1) · 23 Feb 2021

**Response to the Reviewer #1**

This paper presents long-term concentration and isotopic ratio measurements of TC and WSOC in ambient PM2.5 collected at two sites in Japan between July 2017 and July 2019. The authors show a quite impressive series of measurements aiming to investigate sources of the aerosol fine fraction at a suburban and a rural background site. Using stable isotope analyses represents a novel approach for source characterization, thus, is suitable for this goal. The wet oxidation prior to IRMS is very challenging but opens up the opportunity to much easier separate polar compounds and measure their isotopic ratios.
Unfortunately, the presentation is on a poor level. Therefore, it needs to be substantially improved before publishing.
**Response:**
**We thank the reviewer for the insightful comments, which have helped us to significantly improve the paper. We believe through addressing these comments, the quality of the manuscript and its potential impact has been improved. As the reviewer suggested, we discussed sections 3.5.1 and 3.5.2 in greater depth as follow topics.**

**Topic 1. Source identification at suburban site in annual (section 3.5.1)**
**WSOC in every season at suburban site was affected by biomass burning using WSOC concentration, EC concentration non-sea-salt potassium (nss-K$^+$) concentration. In addition, since the C3 and C4 plants have different metabolic pathways, the $\delta^{13}$C values are −32 to −20‰ for C3 plants and −17 to −9‰ for C4 plants, respectively (Smith and Epstein, 1971). The average $\delta^{13}C_{WSOC}$ in suburban site was −25.2 ± 1.1‰, suggesting that biomass burning of C3 plants may be a dominant source. The results were consisted with previous research.**

**Topic 2. Source identification at rural site from autumn to spring and summer (section 3.5.2)**
**WSOC during autumn to spring at rural site was affected by biomass burning using WSOC, EC and non-sea-salt potassium (nss-K$^+$). In particular, biomass burning of C3 plants seemed to be dominant source of WSOC using isotope. In summer, the formation of secondary organic aerosols from biogenic volatile organic compounds (VOCs) was affected to WSOC using biogenic VOCs and isotope fractionation from gas to particle. From February to April 2019, $\delta^{13}C_{WSOC}$ became heavier. The reason seemed to be affected both by biomass burning and aging of OC during long-range transport selectively enriches the $^{13}$C content in organic aerosols, leading to heavier $\delta^{13}$C values in the remaining aerosol.**

**Detailed point by point responses are given below.**

General comments:
Far too little emphasis is placed on the contribution of the isotope measurements to elucidate aerosol sources or chemical processing during atmospheric transport. The main criticism for this manuscript is that the isotopic discussions are generally kept at a very superficial level. The lines of reasoning are often vague, sometime contrived.
**Response:**
**As the reviewer suggested, we discussed sections 3.5.1 and 3.5.2 in greater depth as above topics. In particular, the discussion about C3, C4 plants and long-range transport were added to be interpreted using isotopes. As a result, the dominant annual source for WSOC was C3 plant burning in Tsukuba. In Yurihonjo, the heavy $\delta^{13}C_{WSOC}$ from autumn to spring were a result mainly of biomass burning of rice straw, whereas the light $\delta^{13}C_{WSOC}$ in summer was a result mainly of the formation of secondary organic aerosols from biogenic volatile organic compounds. Thus, our $\delta^{13}C_{WSOC}$ approach was useful to elucidate the sources and atmospheric processes that contribute to seasonal variations of WSOC concentrations.**
**We revised the sections 3.5.1 and 3.5.2.**

Specific comments:

1) The authors present d13C for WSOC and TC. They compare their observations with isotopic ratios of single tracers such as for levoglucosan or toluene, which is quite senseless. As for the former (Sections 3.5.1 and 3.5.2 as well as in the abstract), even during intense biomass burning activities, levoglucosan contributes with few percent to OC and consequently, with less then 10% to the WSOC. Due to the complexity of the biomass burning sources and processes, it is very unlikely that the levoglucosan source specific d13C (please note here that Sang et al. EST2012 presented d13C0 of levoglucosan in aerosol formed during the combustion of C3 plants only) will determine alone the WSOC or TC d13C.

**Response:**

**As the reviewer pointed out, levoglucosan concentration accounts for only about 3.8% of the WSOC concentration in urban area of Japan (Kumagai et al., 2010). Therefore, we think that the $\delta^{13}C$ of only levoglucosan cannot prove that the source of WSOC. As described above topics, we interpreted the results not only for $\delta^{13}C_{TC}$ and $\delta^{13}C_{WSOC}$, but also carbon component and water-soluble ion concentrations. As a result, the dominant annual source for WSOC was C3 plant burning in Tsukuba. In Yurihonjo, the heavy $\delta^{13}C_{WSOC}$ from autumn to spring were a result mainly of biomass burning of rice straw. For $\delta^{13}C$ of levoglucosan ($-25.86 \pm 0.27$‰ to $-24.26 \pm 0.09$‰ (Sang et al., 2012)), the average $\delta^{13}C_{WSOC}$ in Tsukuba was close to the $\delta^{13}C$ of levoglucosan from burning rice straw. Although $\delta^{13}C$ of components other than WSOC may exhibit the similar mechanism as levoglucosan, this interpretation was outside the scope of this paper.**

The arguments used to interpret the isotopically lighter aerosol in summer are laboured, too. The authors cite here Irei et al., who investigated isotopic ratios of different generation reaction products of toluene oxidation. The depletion by 6 permille is valid only for the mentioned reaction (since KIE epsilon is 5.95 permille) and only for its early stages. For other fossil combustion tracers, oxidation reactions show a large range of KIE (Anderson et al. GRL2004), therefore their products will be very differently depleted at the reaction beginning. Compound specific isotope measurements of single tracers are necessary for detailed studies.

**Response:**

**The $\delta^{13}C_{WSOC}$ in summer at Yurihonjo was very light ($-27.4$‰) compared with the average value for the observation period. As described above topic 2, we concluded that the light $\delta^{13}C_{WSOC}$ in summer was a result mainly of the formation of secondary organic aerosols from biogenic VOCs. As VOCs in the atmosphere are oxidized by photochemical oxidants, the $\delta^{13}C$ of residual VOCs becomes heavier by isotopic fractionation (Rudolph et al., 2000; Anderson et al., 2004). In other words, secondary production tends lighter $\delta^{13}C$ of SOA in the atmosphere. Actually, the $\delta^{13}C$ of SOA particles formed by photooxidation of toluene was 3 to 6‰ lighter in laboratory-based experiment than that of the precursor toluene, varying systematically with the extent of the oxidation reaction (Irei et al., 2006; Irei et al., 2011). Assuming this isotope fractionation for toluene applies also to all other potential components, the $\delta^{13}C$ of the anthropogenic VOCs emission source for Yurihonjo was calculated as approximately $-24.4$ to $-21.4$‰ by subtracting 3 to 6‰ from the average $\delta^{13}C_{WSOC}$ during summer in Yurihonjo ($-27.4$‰). This estimated $\delta^{13}C$ value of VOCs was heavier than those previously reported for anthropogenic VOCs. Therefore, the anthropogenic VOCs was considered no dominant source of WSOC. We revised the sentences (page 10, lines 296-304).**

Yet, the TC and WSOC d13C can be compared with corresponding values (Gensch et al. IJMS2014). The observations agree mainly with TC and WSOC d13C of C3 plant and fossil fuel combustion aerosol.

**Response:**

**As described above topics, we concluded that sources of WSOC affected by C3 plants biomass burning and/or aging of OC during long-range transport. And, $\delta^{13}C_{WSOC}$ in summer was a result mainly of the formation of secondary organic aerosols from biogenic volatile organic compounds.**

2) As for the TC and WSOC d13C seasonal trend, the authors state in Lines220-224: *'d13CTC and d13CWSOC at Tsukuba became slightly heavy from February to April 2019, but they showed no other clear seasonal variation (Fig. 2a). In contrast, the d13CTC and d13CWSOC at Yurihonjo were heaver in winter and spring than in summer and autumn (Fig. 2b), and they showed a significant seasonal*

*variation (d13CTC; p < 0.01, d13CWSOC; p < 0.01) compared to those in Tsukuba. At both study sites, d13CWSOC was usually heavier than d13CTC, but in summer d13CWSOC was comparable to or lighter than d13CTC.'*

Firstly, most of the differences between TC and WSOC d13C seem to be within the uncertainty range for isotopic measurements. To show the opposite, the authors should present some statistical evidence.

**Response:**
**The average-1SD for $\delta^{13}C_{TC}$ and $\delta^{13}C_{WSOC}$ was very small at 0.12‰ (<0.46‰) and 0.09‰ (<0.50‰), respectively, for all samples examined in the present study (page 5, lines 131-132).**
**We calculated the differences ($\delta^{13}C_{WSOC}$ - $\delta^{13}C_{TC}$) and performed statistical processing. Both sites showed a statistically significant difference in summer (Tsukuba; $p < 0.01$, Yurihonjo: $p < 0.01$). We added the statistical results in the revised manuscript (page 7, lines 214-215).**

Secondly, the unlike seasonal variation between the two sites give some additional information, which should be discussed in more detail. In absence of compound specific analyses, there are only qualitative indications, but exactly such discussions (e.g. local sources close by the suburban sampling site, which 'flatten' the influence of the long range transport) would enlighten the advantages of using stable isotopes in atmospheric studies. The enrichment by up to 6 permille in winter cannot be explained by chemistry alone, considering the lower oxidant concentration in the cold season. Rather, it looks like significant contribution of heavier sources to the collected aerosol (coal or even C3 plant combustion, see Gensch et al. IJMS2014). This assumption is supported by the similar d13C for TC and WSOC. A back-trajectory analysis would help to elucidate such questions.

**Response:**
**The $\delta^{13}C_{WSOC}$ became heavier at Yurihonjo from February to April 2019. Aerosol photochemical aging during long-range transport selectively enriches the $^{13}C$ content in organic aerosols, leading to heavier $\delta^{13}C$ values in the remaining aerosol (Kirillova et al., 2013a; Bosch et al., 2014; Dasari et al., 2019; Zhang et al., 2019). In a field study, the isotope fractionation values for $\delta^{13}C_{WSOC}$ were estimated to be enriched by 3‰–4‰ because of aging during transport (Kirillova et al., 2013b). Therefore, we speculate that the heavier $\delta^{13}C_{WSOC}$ from winter to spring 2019 at Yurihonjo is affected both by biomass burning and aging of OC during long-term transport. Using Community Multiscale Air Quality model results, Uranishi et al. (2020) concluded that biomass burning in northeast China was transported in Akita prefecture regions of Japan in February and March 2019. We revised the sentences (page 9, lines 275-285).**

3) Line124: remove *'units, were calculated as follows'*. Suggestion: 'Stable carbon isotope ratios are expressed in terms of δ notation in permil (‰:' Equation 1 gives only the meaning of d13C. In the lab, CO2 working standards are used (calibrated against IAEA standards). The final d13C values are reported relatively to the international reference VPDB. The calculations behind are described in Brand et al. PureApplChem2010.

**Response:**
**As the reviewer suggested, we revised the sentences as follows (page 4, line 118).**
**"Stable carbon isotope ratios are expressed in δ notation in permil (‰)"**
**We performed the two-point linear calibration to determine $\delta^{13}C$ (Coplen et al., 2006). The calibration method is described in the revised manuscript (page 4-5, lines 122-132).**

Editorial revisions:
The authors should consider renouncing to mention the used computer OS (Lines107, 122, 123)

**Response:**
**As the reviewer suggested, we deleted the words (page 5, lines 102, 116, 117).**

Generally, the used English is not optimal. I strongly suggest that this manuscript is carefully revised by a native speaker.

**Response:**
**We already had two native speakers check our manuscript before we submitted it.**

Some examples of the inadequate language:

1) Unhandy expressions

Lines38-39: *'Although it is possible to estimate the contribution rate using PMF, it is necessary to identify the characterisation of source artificially.'* What is the meaning of 'artificially'? –

**Response:**

**As the reviewer suggested, we deleted the sentences.**

Line91: ' Every December to February'

**Response:**

**As the reviewer suggested, we changed "Every December to February" to "Every year from December to February" in the revised manuscript (page 3, line 87).**

2) Confusing indications:

Lines157, 159: *'... than the Japan Environmental Standard for the annual average...'* It is surely meant a threshold stipulated in the air quality guidelines of the Japan Environmental...organisation.

**Response:**

**As the reviewer suggested, we changed "the Japan Environmental Standard for the annual average" to "the air quality standard for the annual average of Japan" in the revised manuscript (page 5, lines 151, 153).**

3) Wrong wording

Replace 'reasonable' describing the measurements (lines171, 186). Use instead 'good agreement', 'similar to other studies', 'as expected'...

**Response:**

**As the reviewer suggested, we revised the sentences as follows.**

**"The mass concentration of PM$_{2.5}$ were consistent with the seasonal variation for nearby sites of Atmospheric Environmental Regional Observation System (AEROS) provided by the Ministry of the Environment (Ministry of the Environment, 2021)." (page 6, lines 161-163).**

**"The higher percentage contribution to the PM$_{2.5}$ mass concentration from OC than EC was in agreement with compared to other studies" (page 6, lines 176-178).**

Lines226-227: Rephrase: *'d13CWSOC in TSP in Seoul, South Korea, from March 2015 to January 2016 showed no seasonal variation (Han et al., 2020), which is comparable with our present findings for the suburban site, Tsukuba.'* The findings of this work should be presented first and then compare them with other studies.

**Response:**

**As the reviewer suggested, we revised the sentences as follows (page 7, lines 217-220).**

[revised manuscript text omitted]

---

## Author Comment (AC2) · 23 Feb 2021

**Response to the Reviewer #2**

The manuscript entitled "Measurement report: Source characteristics of water-soluble organic carbon in PM2.5 at two sites in Japan, as assessed by long-term observation and stable carbon isotope ratio" by N. Suto and H. Kawashima represents investigation of aerosol particles through the prism of carbon stable isotopes, EC, OC and ion analysis. The sample collection was performed during two years, which allowed for the authors to catch the seasonal changes in some aerosol parameters.

In general, the manuscript is a bit chaotic, sometimes pointing at not relevant issues. Also, I think Authors have not exploited the benefits of long time samples collection and analysis.

I would like to address the issues that I am concerned on:

**Response:**
**We thank the reviewer for the insightful comments, which have helped us to significantly improve the paper. We believe through addressing these comments, the quality of the manuscript and its potential impact has been improved. As the reviewer suggested, we discussed sections 3.5.1 and 3.5.2 in greater depth as follow topics.**

**Topic 1. Source identification at suburban site in annual (section 3.5.1)**
**WSOC in every season at suburban site was affected by biomass burning using WSOC concentration, EC concentration non-sea-salt potassium (nss-K$^+$) concentration. In addition, since the C3 and C4 plants have different metabolic pathways, the δ$^{13}$C values are −32 to −20‰ for C3 plants and −17 to −9‰ for C4 plants, respectively (Smith and Epstein, 1971). The average δ$^{13}$C$_{WSOC}$ in suburban site was −25.2 ± 1.1‰, suggesting that biomass burning of C3 plants may be a dominant source. The results were consisted with previous research.**

**Topic 2. Source identification at rural site from autumn to spring, and summer (section 3.5.2)**
**WSOC during autumn to spring at rural site was affected by biomass burning using WSOC, EC and non-sea-salt potassium (nss-K$^+$). In particular, biomass burning of C3 plants seemed to be dominant source of WSOC using isotope. In summer, the formation of secondary organic aerosols from biogenic volatile organic compounds (VOCs) was affected to WSOC using biogenic VOCs and isotope fractionation from gas to particle. From February to April 2019, δ$^{13}$C$_{WSOC}$ became heavier. The reason seemed to be affected both by biomass burning and aging of OC during long-range transport selectively enriches the $^{13}$C content in organic aerosols, leading to heavier δ$^{13}$C values in the remaining aerosol.**

**Detailed point by point responses are given below.**

Lines 25-30 In the introduction part, the most recent reference is from the year 2011.
**Response:**
**As the reviewer suggested, we added the latest references (Du et al., 2014; Zhang et al., 2019) (page 1-2, lines 28-30).**

Line 35. Why authors mention about PMF? They are not using PMF in this manuscript? What it means "it is necessary to identify the characterisation of source artificially"?
**Response:**
**We don't use PMF models. As the reviewer suggested, we deleted the sentences.**

Line 45. In general, the isotope ratio literature review part is too narrow in this manuscript. I propose to make overview on most recent references, which deals with aerosols and stable isotopes in different areas, which are relevant for this manuscript: transportation, natural terrestrial and marine (also oceanic) sources, fires etc. Suggestions: Zhang W. et al., 2019, ACP 19, 11071-11087 (δ13C of WSOC in China); Fisseha R. et al 2006, Atmos. Env. 43 431-437 (δ13C values in the different aerosol fractions in Zurich, Switzerland); Kirillova E.N. et al. 2010 Anal. Chem. 82 7973-7978; Lang S.Q. et al. 2012 Rapid Commun Mass Sp. 26 9-16; Suto N. and Kawashima H. Rapid Commun Mass Sp. 32 1668-1674 (In this manuscript the same Authors present the own developed method for δ13C in WSOC). The last citation is used later, but I propose to cite it here also, as it is relevant to this part.

**Response:**
**As the reviewer suggested, we revised the isotope part of the introduction follows (page 2, lines 42-57).**
**"The stable carbon isotope ratio ($\delta^{13}C$) of carbonaceous aerosols can provide useful information about a sample of PM (e.g., Widory et al., 2004; Fisseha et al., 2009; Cao et al., 2011; Gensch et al., 2014). Since EC is unreactive, it is possible to identify the source from $\delta^{13}C_{EC}$ in their aerosols directly (e.g., Kawashima and Haneishi, 2012; Zhao et al., 2018). In contrast, OC reacts in the atmosphere, so their $\delta^{13}C_{OC}$ provide information not only on the source of the PM but also on any atmospheric processing it has undergone (e.g., Cao et al., 2011; Ni et al., 2018). The measurement of $\delta^{13}C_{WSOC}$ in PM has been actively carried out in recent years (e.g., Kirillova et al., 2010; Kirillova et al., 2013; Suto and Kawashima, 2018; Zhang et al., 2019). The analysis of $\delta^{13}C_{WSOC}$ in ambient aerosol has been performed by wet oxidation method using GasBench/isotope ratio mass spectrometry (IRMS) (Fisseha et al., 2006) and combustion method using elemental analyzer/IRMS (EA/IRMS) (Kirillova et al., 2010). In the past few years, a highly sensitive analytical methods for $\delta^{13}C_{WSOC}$ based on wet oxidation using liquid chromatography/IRMS (LC/IRMS) (Suto and Kawashima, 2018) and GasBench/IRMS (Zhang et al., 2019), and total organic carbon analyzer/IRMS (Han et al., 2020) have been developed. The combustion method is the most widely used approach today. The $\delta^{13}C_{WSOC}$ of various particle size collected has been reported at various times in East Asia (Miyazaki et al., 2012; Kirillova et al., 2014a; Pavuluri and Kawamura, 2017; Yan et al., 2017; Suto and Kawashima, 2018; Zhang et al., 2019; Han et al., 2020), South Asia (Kirillova et al., 2013; Bosch et al., 2014; Kirillova et al., 2014b; Dasari et al., 2019), Europe (Fisseha et al., 2006; Fisseha et al., 2009; Kirillova et al., 2010), and the United States (Wozniak et al., 2012a; Wozniak et al., 2012b) (Table S1 in the Supplement)."**

Section 2.1. I see that present study two sampling points were separated by a distance of 370 km . Both Tsukuba and Yurihonjo has a Humid continental climate, with different annual mean temperature of about 2∘C. Tsukuba is close to Ocean, while Yurihonjo is close to Sea of Japan. I assume that air mass back trajectories are important when comparing two sampling points in this case. I assume that can be problematic to compare suburban or rural places, as they can be influenced by totally different atmosphere (and aerosol sources) conditions. In the rest of the text I'm missing marine source evaluation (except nss-K+).
**Response:**
**We considered that we need to compare the two sampling sites (suburban site and rural site). The suburban site is located in the inland Kanto plain approximately 60 km northeast of the Tokyo metropolitan area. This site is surrounded by residential areas and forests, and there is a road in front of the building. In contrast, the rural site faces the Sea of Japan, and had no local pollutant sources such as large factories. In particular, reports of $\delta^{13}C_{WSOC}$ in rural site are limited, so we believe that this study will be a very important result.**

Line 95. I did not understand how long sampling for one filter was performed, does all filters were loaded periodically? Maybe worth to add as supplementary filter data (time, weight of loaded aerosol). How Authors ensured that the filter not lost mass itself, as quartz filters break easily, and making gravimetric measurements can be complicated. Does Authors used equilibration for the unexposed and exposed filter to account for humidity?
**Response:**
**PM$_{2.5}$ samples were collected approximately every 10 days in Tsukuba and 14 days in Yurihonjo (page 3, lines 84, 88) in order to obtain enough carbon amount in samples for isotope analysis.**
**We confirmed that the PM$_{2.5}$ mass concentration calculated from the filter weight was in agreement with the result of the automatic measuring instrument near the sampling site (Ministry of the Environment, 2021). The slope of the scatter plot was 1.27 for Tsukuba and 0.95 for Yurihonjo. Thus, we think the filter weight is reasonable.**

Section 2.2 can be improved. I miss a clear step-by-step description of analytical procedures: Line 105: ". . .the carbon was converted to CO2 via an oxidation catalyst in the reduction tube of EA" - I think is incorrect and must be clarified. Does oxygen was added? Does nitrogen was seperated from CO2? What

about blank measurements, does it gave any signal? In this section, is no clear separation, which part of the section corresponds to total carbon measurements, which part to WSOC?

**Response:**

**The analytical methods for $\delta^{13}C_{TC}$ and $\delta^{13}C_{WSOC}$ were described lines 96-102 and lines 103-115, respectively.**

**As the reviewer suggested, we revised the part of the measurement for $\delta^{13}C_{TC}$ using EA/IRMS (page 4, lines 96-102). No filter blank was detected.**

Line 170. Authors state that snow is responsible for low PM2.5 concentration in Yurihonjo. My question, does land in winter emits significant amount of aerosol, even not covered by snow? What about heating activities, long range transport etc. I want to say that Authors must to think generally what they want to deliver to the reader, and maybe skip parts which are not relevant to the main purpose of the article (for example about most polluted megacities, as here megacities are not investigated). Yurihonjo can't be compared to a megacity, as by definition it is an urban site (last line in 165).

**Response:**

**As the reviewer pointed out, we couldn't find any evidence that snow is the reason for low PM$_{2.5}$ concentration. Therefore, we deleted this sentence. As described topic 2, we revised the section 3.5.2.**

**We think that information on PM$_{2.5}$ concentrations in global megacities is necessary for this manuscript. This part shows that PM$_{2.5}$ concentration in this study is low enough compared to the polluted megacities (page 5-6, lines 155-160).**

Line 260-265. The assumption that rice residues burning is the main source of WSOC in Tsukuma is nice assumption, but I was not convinced by this statement. Usually, in urban sites weak seasonal trend in carbon delta values is due to transportation influence. Here, Authors did not evaluate the $\delta13C$ of fossil fuel emissions. In addition, assuming that crop combustion is the main source of WSOC, we must to see correlation with delta values. No correlation between WSOC concentrations and delta values means that does not exist one dominating aerosol source. In other words, if in Tsukuma main source would be crop burning, we must observe correlation between WSOC and delta values. Or, delta values are affected by atmospheric processing, but then air mass back trajectory analysis (even with Solar intensity) must be performed to understand the resident time of the aerosol.

**Response:**

**The WSOC have both primary and secondary sources that can be biogenic or anthropogenic. Also biomass burning is another important source of WSOC (Sullivan and Weber, 2006). As described topic 1, the $\delta^{13}C_{WSOC}$ was consistent with the $\delta^{13}C$ of C3 plant, and a high correlation was found between WSOC concentration and non-sea-salt potassium concentration in Tsukuba. These results suggested that the main source of WSOC was biomass burning of rice straw. In fact, the season of biomass burning for rice straw is from September to October in Tsukuba (Tomiyama et al., 2017).**

**As the reviewer pointed out, a weak correlation ($r = 0.18$) was found between $\delta^{13}C_{WSOC}$ and WSOC concentrations. If the value of $\delta^{13}C$ is constant, the correlation between $\delta^{13}C$ and concentration will be low.**

Line 290. Speculations about long-range transport must be verified by air mass trajectory analysis. Also, Authors can use other delta signatures (various crops, land and forest fires, transportation and fuel burning emissions, shipping emissions etc.), not only rise burning.

**Response:**

**The result of seasonal back trajectory analysis at Yurihonjo has been described in Kawashima and Haneishi (2012). The seasonal variations of air masses show similar trends every year. The air masses came mainly from northeastern China and Siberia during autumn and spring, while they came from the Pacific Ocean in summer.**

**Since there are no $\delta^{13}C_{WSOC}$ data from the various sources (e.g., fossil fuel, transportation, and fuel burning) pointed out by the reviewers, we could not be compared in this study.**

Line 310. Kawashima and Murakami 2014 report that in roadside samples VOC ranged from -29.6 to 23.5 permill (the same range for ambient samples). It makes difficulty interpreting vehicular emissions from a stable isotopes perspective. I'm not sure if only Toluene and Xylene are responsible for WSOC in Yurihonjo, so 5.8 permill subtraction must be made carefully (Authors note themselves that anthropogenic emissions of VOC are smaller than that biogenic).

**Response:**

**As the reviewer suggested, it is difficult to estimate the effects of anthropogenic VOCs and biogenic VOCs of as sources of WSOC. In particular, WSOC are formed by oxidation of VOCs and involve isotope fractionation. This study considered the effects of anthropogenic VOCs using the isotope fractionation from VOCs to SOA. As a result, the anthropogenic VOCs was considered no dominant source of WSOC. Therefore, we speculated that the light $\delta^{13}C_{WSOC}$ in summer was a result mainly of the formation of SOA from biogenic VOCs.**

**We revised the sentences to section 3.5.2 (page 10, lines 286-304).**

Figure 2. I not found in the manuscript explanation, what causes enriched delta values in the winter in season 2018/2019 compared to 2017/2018 at both locations?

**Response:**

**The $\delta^{13}C_{WSOC}$ from February to April 2019 was heavy at both sampling sites. The reason is affected both by biomass burning and aging of OC during long-term transport in Yurihonjo. In contrast, the reason for the heavy $\delta^{13}C_{WSOC}$ in Tsukuba remains unclear.**

**We added the sentences to section 3.5.2 (page 9, lines 275-285).**

[revised manuscript text omitted]

---

## Author Response (AR2)

**Response to the Reviewer #1**

Measurement report: Source characteristics of water-soluble organic carbon in PM2.5 at two sites in Japan, as assessed by long-term observation and stable carbon isotope ratio by Suto and Kawashima
The revised manuscript shows that the authors considered only some of the recommendations suggested by the reviewers. Therefore, there are still open points to be reconsidered.
**Response:**
**We thank the reviewer for the insightful comments, which have helped us to significantly improve the paper. We believe through addressing these comments, the quality of the manuscript and its potential impact has been improved. Detailed point by point responses are given below.**

The major criticism was unfortunately not reviewed: since only TC isotopic ratios of WSOC and whole aerosol particles were measured, comparison with compound specific d13C are superfluous (to illustrate this, two extremes should be considered, i.e. 3.8% levoglucosan originates from the combustion of corn plant only (d13C=-12 permille) or a C3 wood only (mean d13C=-24 permille). Keeping the rest unchanged, a difference in the bulk d13C of 0.46 permille would result, which is in the uncertainty range of some compound specific analyses). The same is valid for process examination. One of prerequisites to employ mass balance calculations for single mixture components /photo chemical decay is to possess their emissions d13C and the reaction specific KIE. Even then, the complexity of atmospheric processing cannot be solved by just adding up some numbers.
**Response:**
**In our manuscript, the source of WSOC is discussed using $\delta^{13}C_{WSOC}$, carbon components (EC and WSOC concentrations), and water-soluble ion (nss-$K^+$ concentration) rather than the comparison of compound specific. As the reviewer pointed out, it is not possible to directly compare $\delta^{13}C_{WSOC}$ and $\delta^{13}C$ of levoglucosan due to very low percentages. However, the agreement between the average $\delta^{13}C_{WSOC}$ in Tsukuba (−25.2 ± 1.1‰) and $\delta^{13}C$ of levoglucosan for rice straw (−24.26 ± 0.09‰; Sang et al. (2012)) seemed to be important information. Only $\delta^{13}C$ of levoglucosan has been reported at present, it should be considered to investigate the $\delta^{13}C$ of individual components not only levoglucosan in the future research.**
**We revised the discussion in section 3.5.1 as follows (page 9, lines 264-271).**
**"The main chemical component generated by the breakdown of cellulose by burning rice straw is levoglucosan, which can be used as a tracer of biomass burning (Simoneit et al., 1999). The $\delta^{13}C$ of levoglucosan emitted from burning rice straw, peanut stalk, mulberry stalk, China fir, Chinese red pine, chinese guger tree, Chestnut such as C3 plants ranged from −26.05‰ to −22.60‰, especially from rice straw, which was −24.26 ± 0.09‰ (Sang et al., 2012). The average $\delta^{13}C_{WSOC}$ in Tsukuba was very close to the $\delta^{13}C$ of levoglucosan from burning rice straw. However levoglucosan concentration accounts for only about 3.8% of the WSOC concentration in urban area of Japan (Kumagai et al., 2010), and was very low percentages. Thus, it is difficult to compare the source directly using only the $\delta^{13}C$ of levoglucosan. In the future research, it should be considered to investigate the $\delta^{13}C$ of individual components not only levoglucosan."**

Yet, there are numerous ambient studies on WSOC/TC isotope ratios (see Gensch et al. IJMS2014 and references therein), describing ranges for these compound classes. A sound comparison gives first information on sources. The authors claim that: *'When C3 plants are burned in the laboratory, there is no significant δ13C difference between the produced particles and original C3 plants'*. Still, isn't sounder to compare the measurements with WSOC/TC d13C in combustion aerosol particles and not in plant tissues (whose δ13C values are −34 to −24‰ for C3 plants and −19 to −6‰ for C4 plants, respectively (Smith and Epstein, 1971) and not −32 to −20‰ for C3 plants and −17 to −9‰ for C4 plants, as given in the present manuscript), since this information already exists?
**Response:**
**As the reviewer suggested, we added $\delta^{13}C$ values from C3 plants and C4 plants burning (Kawashima and Haneishi, 2012; Garbaras et al., 2015; Guo et al., 2016). And, we revised the $\delta^{13}C$ of original C3 and C4 plants in the reference (Smith and Epstein, 1971) .**
**We revised the discussion in section 3.5.1 as follows (page 8-9, lines 250-257).**
**"The average $\delta^{13}C_{WSOC}$ was −25.2 ± 1.1‰ in Tsukuba. Since the C3 and C4 plants have different**

metabolic pathways, the $\delta^{13}C$ values are −34 to −24‰ for C3 plants and −19 to −6‰ for C4 plants, respectively (Smith and Epstein, 1971). When C3 plants are burned in the laboratory, there is no significant $\delta^{13}C$ difference between the produced particles and original C3 plants (Turekian et al., 1998; Das et al., 2010). In contrast, the particles produced by burning the C4 plants were 3.5‰ lighter than the original C4 plants (Turekian et al., 1998). Therefore, the $\delta^{13}C$ of C4 plants was estimated to be −22.5 to −9.5‰. In fact, the $\delta^{13}C$ from C3 plants and C4 plants burning were −34.7 to −25.1‰ and −19.3 to −16.1‰, respectively (Kawashima and Haneishi, 2012; Garbaras et al., 2015; Guo et al., 2016). Thus, the average $\delta^{13}C_{WSOC}$ at Tsukuba suggested that biomass burning of C3 plants might be a dominant source."

1) To get some additional information from this impressive dataset, I recommend to the authors to use the existing isotopic tools, such as combining isotope ratios with concentration data (remote sources for Yurihonjo, higher contribution of local/regional emissions to the samples for Tsukuba?). The authors claim in Lines 275-279: *'From February to April 2019, d13CWSOC became heavier with increasing WSOC concentrations (Fig. 1b and Fig. 2b). Although C4 plants such as corn and grass showed heavy d13C, there was no evidence of burning of C4 plants around during this period. The heavy d13CWSOC during this period cannot be explained only by the effect of biomass burning. Aerosol photochemical aging during long-range transport selectively enriches the 13C content in organic aerosols, leading to heavier d13C values in the remaining aerosol'.* A simultaneous increase in concentration and d13C is rather an indication for heavy sources than aging... especially during periods of low photo chemical activity.

**Response:**

As the reviewers pointed out, $\delta^{13}C_{WSOC}$ in Yurihonjo was the heaviest with increasing WSOC concentrations (average, $1.5 \pm 0.7$ µg m$^{-3}$; −21.3 ± 1.9‰) from February to April 2019 (Fig. 1b and Fig. 2b). This $\delta^{13}C_{WSOC}$ value might be related to heavy $\delta^{13}C$ source such as C4 plants (e.g., corn and grass). Around the sampling site at Yurihonjo, there was no evidence of burning of C4 plants during this period. We investigated the fire spots using Fire Information for Resource Management System (FIRMS) (NASA, 2017) and air mass backward trajectories using the National Oceanic and Atmospheric Administration (NOAA) Hybrid Single-Particle Lagrangian Integrated Trajectory (HYSPLIT) model (Draxler and Rolph, 2013). According to the fire spots data, the number of fire spots were observed during this period in Northeast China (Fig. 1 in this Response file / Fig. S2 in the Supplement). Northeast China is the largest producer of corn in China (MWCACP, 2019), and biomass burning is actively used for heating in winter (Chen et al., 2017). The air mass backward trajectories showed that air masses during this period at Yurihonjo were mainly derived from areas located northeast China (Fig. 2 in this Response file / Fig. S3 in the Supplement). Recently, aerosol photochemical aging during long-range transport selectively enriches the $^{13}C$ content in organic aerosols, leading to heavier $\delta^{13}C$ values in the remaining aerosol (Kirillova et al., 2013a; Bosch et al., 2014; Dasari et al., 2019; Zhang et al., 2019). Since it could not be denied that aging effect would not occur from February to April 2019 in Yurihonjo, we speculate that the heavier $\delta^{13}C_{WSOC}$ might be affected by C4 plant burning and/or aging during long-term transport.

We revised the discussion in section 3.5.2 as follows (page 9-10, lines 281-297).

"From February to April 2019, $\delta^{13}C_{WSOC}$ was the heaviest with increasing WSOC concentrations (average, $1.5 \pm 0.7$ µg m$^{-3}$; −21.3 ± 1.9‰) (Fig. 1b and Fig. 2b). The moderate correlation between $\delta^{13}C_{WSOC}$ and WSOC concentration was observed ($r = 0.54$, $p = 0.27 > 0.1$). This $\delta^{13}C_{WSOC}$ value might be related to heavy $\delta^{13}C$ source such as C4 plants (e.g., corn and grass). Around the sampling site at Yurihonjo, there was no evidence of burning of C4 plants during this period. Northeast China is the largest producer of corn in China (MWCACP, 2019), and biomass burning is actively used for heating in winter (Chen et al., 2017). In Figure S2 in the Supplement, the number of fire spots were observed from February to April 2019 (NASA, 2017). The air mass backward trajectories showed that air masses during this period at Yurihonjo were mainly derived from areas located northeast China (Fig. S3 in the Supplement). For other air model, Uranishi et al. (2020) concluded that biomass burning in northeast China was transported in Akita prefecture regions of Japan in February and March 2019 from Community Multiscale Air Quality model results. For water-soluble ion data, the correlation between Na$^+$ and Cl$^-$ concentration was

**highest from winter to spring 2019 ($r = 0.98$, $p < 0.01$), suggesting the influence of sea salt. Recently, aerosol photochemical aging during long-range transport selectively enriches the $^{13}$C content in organic aerosols, leading to heavier $\delta^{13}$C values in the remaining aerosol (Kirillova et al., 2013a; Bosch et al., 2014; Dasari et al., 2019; Zhang et al., 2019). In a field study, the isotope fractionation values for $\delta^{13}C_{WSOC}$ were estimated to be enriched by 3‰–4‰ because of aging during transport (Kirillova et al., 2013b). We speculate that the heavier $\delta^{13}C_{WSOC}$ from February to April 2019 at Yurihonjo might be affected by C4 plant burning and/or aging during long-term transport."**

2) To either prove or reject this hypothesis, the authors should at least roughly (HYSPLIT?) determine the origin of the air masses. In the lines 282-284 the authors state: *'Uranishi et al. (2020) concluded that biomass burning in northeast China was transported in Akita prefecture regions of Japan in February and March 2019 from Community Multiscale Air Quality model results'*. Noticeably here, China's north-eastern part is belonging to the so called 'corn belt'. As a result of this farming type, it is expected that local heating with corn briquettes and traditional cooking using corn cobs strongly contribute to the aerosol burden. Subsequently, due to long-range transport, this would contribute to the investigated samples, leading to an increase in d13C.

**Response:**

**As the described above, we investigated the fire spots using Fire Information for Resource Management System (FIRMS) (NASA, 2017) and air mass backward trajectories using the National Oceanic and Atmospheric Administration (NOAA) Hybrid Single-Particle Lagrangian Integrated Trajectory (HYSPLIT) model (Draxler and Rolph, 2013). According to the fire spots data, the number of fire spots were observed during this period in Northeast China (Fig. 1 in this Response file / Fig. S2 in the Supplement). Northeast China is the largest producer of corn in China (MWCACP, 2019), and biomass burning is actively used for heating in winter (Chen et al., 2017). The air mass backward trajectories showed that air masses during this period at Yurihonjo were mainly derived from areas located northeast China (Fig. 2 in this Response file / Fig. S3 in the Supplement).**

**We added the figures of fire spots (Fig. S2 in the Supplement) and backward trajectories (Fig. S3 in the Supplement), and revised the discussion in section 3.5.2 as above (page 9-10, lines 281-297).**

[Figure]

**Figure. 1 Monthly fire spots from July 2017 to June 2019 as determined by MODIS in Fire Information for Resource Management System (FIRMS) (NASA, 2017).**

[Figure]

**Figure. 2 Monthly air masses backward trajectories frequency at Yurihonjo from May 2018 to April 2019 (Draxler and Rolph, 2013).**

**Response to the Reviewer #2**

The manuscript has been improved. I still have few minor remarks.
**Response:**
**We thank the reviewer for the insightful comments, which have helped us to significantly improve the paper. We believe through addressing these comments, the quality of the manuscript and its potential impact has been improved. Detailed point by point responses are given below.**

In Introduction section: *"The wet oxidation/IRMS method described above ... the total analysis time is markedly reduced compared with the combustion method"*. I do not agree with the statement that analysis time for wet oxidation method is shorted compared to the combustion method. Authors detailed described two methods in Section 2.2 Stable carbon isotope analysis, and we see that chemical extraction consumed time for each sample to prepare in wet oxidation method, while for TC combustion analysis time is shorter for one sample.
**Response:**
**This sentence (page 3, lines 70-72) compares the analysis time, including pretreatment time, for the wet oxidation method and combustion method for measuring $\delta^{13}C_{WSOC}$. For the wet oxidation/IRMS method, the sample filter is extracted for 15 min and the extract is measured directly using LC/IRMS (Suto and Kawashima, 2018). On the other hand, the combustion method requires a complex pretreatment process, such as freeze-drying of the aerosol extract under vacuum for 16 hours (Kirillova et al., 2010).**
**As the reviewer suggested, we added the sentences to clarify as follows (page 3, lines 69-70).**
**"The combustion method, which is widely used at present, requires more pretreatment time because samples of PM are extracted, dehydrated with a freeze drier, dried, and then measured by EA/IRMS."**

The sentence *"In addition, this newer approach is highly sensitive, so only small amounts of sample are needed"*. In my opinion, this sentence is misleading, because for wet oxidation method, 14.13cm$^2$ of filter was used, comparing to I guess 1 cm$^2$ for d13C TC. Also, what is mean small amounts? I suggest to use the words "small amount comparing to (another analysis type)". Bear in mind, that here Authors use Hi-Vol sampler of 10 days sampling for one filter with flow rate 1000 l/min, which make high amount of aerosol to accumulate on filter despite sampling location.
**Response:**
**The detection limit of the combustion method for measuring $\delta^{13}C_{WSOC}$ was 120-150 µg of WSOC (Kirillova et al., 2014a; Kirillova et al., 2014b), but the detection limit of the wet oxidation/IRMS method was improved to 1 µgC (Suto and Kawashima, 2018) with high sensitivity. The establishment of the highly sensitive $\delta^{13}C_{WSOC}$ method has made it possible to measure $\delta^{13}C_{WSOC}$ even at sampling sites where the WSOC concentration is relatively low, such as in this study. In addition, $\delta^{13}C_{WSOC}$ measurement with higher time resolution is possible at sampling sites with high WSOC concentration.**
**As the reviewer suggested, we revised the sentences to clarify as follows (page 3, lines 72-73).**
**"In addition, this newer approach is highly sensitive, so only small amounts of sample are needed compared to the combustion method."**

**References**

[revised manuscript text omitted]

---

## Author Response (AR3)

The revised manuscript shows that the authors well considered the recommendations suggested by the reviewers. Yet, there are two more issues:

**Response:**

**We thank the reviewer for the insightful comments, which have helped us to significantly improve the paper. We believe through addressing these comments, the quality of the manuscript and its potential impact has been improved. Detailed point by point responses are given below.**

- Lines296-297: The conclusion of C4 contribution in Feb-Apr 2019 should be emphasized. Instead of 'We speculate that the heavier $\delta^{13}C_{WSOC}$ from February to April 2019 at Yurihonjo might be affected by C4 plant burning and/or aging during long-term transport.'

use something like:

'The combination of isotopic ratio and concentration measurements (Figures 1 and 2) with the prevailing biomass burning activities (Figure S2) and back trajectory analyses (Figure S3) hints that the heavier $\delta^{13}C_{WSOC}$ from February to April 2019 at Yurihonjo was rather caused by contribution of C4 plant combustion than by aging during long-range transport."

**Response:**

**As the reviewer suggested, we revised the sentences to clarify (page 10, lines 300-303).**

- I still recommend a revision by a native English speaker to improve the readability of the manuscript.

**Response:**

**Our paper was edited again by English-speaking professional editors of ELSS, Inc. (elss@elss.co.jp)**